# Nonvolatile nuclear spin memory enables sensor-unlimited nanoscale spectroscopy of small spin clusters

Matthias Pfender [1], Nabeel Aslam[1], Hitoshi Sumiya[2], Shinobu Onoda[3], Philipp Neumann [1], Junichi Isoya [4], Carlos A. Meriles[5] & Jörg Wrachtrup[1]

In nanoscale metrology, dissipation of the sensor limits its performance. Strong dissipation has a negative impact on sensitivity, and sensor–target interaction even causes relaxation or dephasing of the latter. The weak dissipation of nitrogen-vacancy (NV) sensors in room temperature diamond enables detection of individual target nuclear spins, yet limits the spectral resolution of nuclear magnetic resonance (NMR) spectroscopy to several hundred Hertz, which typically prevents molecular recognition. Here, we use the NV intrinsic nuclear spin as a nonvolatile classical memory to store NMR information, while suppressing sensor back-action on the target using controlled decoupling of sensor, memory, and target. We demonstrate memory lifetimes up to 4 min and apply measurement and decoupling protocols, which exploit such memories efficiently. Our universal NV-based sensor device records single-spin NMR spectra with 13 Hz resolution at room temperature.

[1] 3. Physikalisches Institut, University of Stuttgart, Pfaffenwaldring 57, 70569 Stuttgart, Germany. [2] Sumitomo Electric Industries Ltd., Itami 664-0016, Japan. [3] Takasaki Advanced Radiation Research Institute, National Institutes for Quantum and Radiological Science and Technology, Takasaki 370-1292, Japan. [4] Research Center for Knowledge Communities, University of Tsukuba, Tsukuba 305-8550, Japan. [5] Department of Physics, CUNY—City College of New York, 160 Convent Avenue, New York, NY 10031, USA. Correspondence and requests for materials should be addressed to M.P. (email: m.pfender@physik.uni-stuttgart.de) or to P.N. (email: p.neumann@physik.uni-stuttgart.de)

The nitrogen-vacancy (NV) center in diamond has emerged as an exceptional nanoscale, room temperature quantum sensor for magnetic and electric fields, temperature, and pressure[1–6]. These sensor capabilities arise from corresponding susceptibilities of the ground state $S = 1$ electron spin (below called sensor spin) that can be initialized and read out optically[7]. In particular, it has been demonstrated that a single NV center can detect proximal nuclear spins inside and outside of the diamond lattice[8–10]. When coherently coupled to such spins, an NV center constitutes a small quantum processor[11–14]. Furthermore, proton spin ensembles from a nanoscopic volume can be detected[15–17] and distinguished from fluorine or silicon nuclear spins via their distinct gyromagnetic ratios, even at room temperature[10, 18–20]. Although strongly coupled nuclear spins such as the NV center's intrinsic nitrogen and $^{13}$C are excellent candidates for quantum bits[11, 14, 21], weaker coupled spins are rather regarded as bath spins responsible for sensor dephasing[22, 23], though with some potential for quantum simulation with tailored spin baths[24]. NV centers enable identification of such target spins with spectral linewidths of several hundred Hz at room temperature[9, 25], yet, they hamper achieving high-resolution NMR spectroscopy (e.g., chemical shift, J-coupling, few Hz), necessary for structure determination.

The limiting factor for resolution of NV-based nanoscale NMR spectroscopy at room temperature is given by

$$\Delta \nu = \frac{1}{\pi} \left( \frac{1}{T_2^*} + \frac{1}{T^{\text{mem}}} \right), \tag{1}$$

with the target spin-coherence time $T_2^*$ and a sensor-related memory lifetime $T^{\text{mem}}$ in an underlying in situ correlation spectroscopy[9, 25, 26] as will be discussed below (see Table 1 for times scales).

In previous NV-based NMR experiments, the achievable linewidth was limited by the finite relaxation time of the sensor $T_1^{\text{sens}}$ to about $\Delta \nu = \frac{5}{3\pi T_1^{\text{sens}}} \approx 100\,\text{Hz}$[25, 26] (see "Methods"). There are two causes for this limitation. First, at room temperature the sensor-spin relaxation processes establishes a decay channel between its environment and the target, limiting the $T_2^*$ of the latter (first term in Eq. (1)). Second, metrology information about the target spin is irreversibly lost on the $T_1^{\text{sens}}$ timescale, which limits the interrogation or memory lifetime and hence the spectral resolution of the sensing device (second term in Eq. (1)).

Under cryogenic conditions $T_1^{\text{sens}}$ can be regarded as infinite[27] compared with typical sensing times (~100 s vs. <1 s)[28]. At room temperature, however, relaxation times are shorter and we have to resort to other solutions. One example was demonstrated in ref. [21], where high-intensity laser illumination decoupled sensor and a particular target spin. Here, we seek for a universally applicable spectroscopy method for NV sensor systems, which works for a broad range of targets, features fast and efficient measurements, and is robust for a broad range of sensor–target decoupling methods. To this end, we investigate the application of the intrinsic nitrogen nuclear spin (below called memory spin) for robust intermediate storage of classical metrology information. The nonvolatile classical memory furthermore allows decoupling of target spins from the sensor thus closing the decay channel. Common methods are

dynamical decoupling acting on the target spins or by averaging the interaction between sensor and target spins, either by coherent manipulation or stochastic processes acting on the sensor (e.g., pulsed or continuous dynamical decoupling[29], or motional narrowing[21]). Here, we chose two distinct stochastic processes, an active one relying on continuous, low-intensity optical illumination, and a passive one exploiting intrinsic dissipation mechanisms in another charge state (i.e., NV$^0$). Eventually, we retain metrology information on timescales longer than the decay of the sensor spin, which results in Hertz spectral resolution of the target spin resonances.

## Results

**In situ correlation spectroscopy of individual spins.** Small spin ensembles under ambient conditions are typically in a mixed thermal state. A common method for characterizing such target spins is (non-in situ) correlation spectroscopy comprising an initial measurement of their state, followed by a suitable free evolution and a final measurement[21, 28]. When the measurements are strong, they project the target spins, and they yield maximum information for a readout fidelity approaching unity[30]. Therefore, subsequent measurement results are strongly correlated by the intermediate free evolution of the target spins (e.g., Ramsey oscillation). Although this scenario is feasible for NV-based NMR spectroscopy at cryogenic temperatures[28], it is very inefficient at room temperature operation, where a single readout of the NV sensor spin yields negligible information due to insufficient readout fidelity[21, 31, 32]. Furthermore, at room temperature, optical sensor readout might disturb the target spins[33–35] and scrambles the sensor spin itself via ionization thereby reducing the correlation of subsequent readouts[36, 37].

In situ correlation spectroscopy circumvents all three challenges at room temperature. It consists of two phase-accumulation parts with total duration $\tau < T_2^{\text{sens}}$ representing initial and final measurement, separated by a correlation time $T_c < T_1^{\text{sens}} \gg T_2^{\text{sens}}$ (Table 1)[9, 26, 38–40]. The initial state of the target spins, encoded in the first accumulated phase, however, is not readout but stored on a memory. The sensor-spin expectation value $\langle S_z \rangle$ is often used as a memory because its (longitudinal) relaxation time $T_1^{\text{sens}} \approx 6\,\text{ms}$ is typically one to three orders of magnitude longer than its transverse relaxation time $T_2^{\text{sens}}$ (Fig. 1b). During $T_c$, the target spins may be manipulated, for example, by a Ramsey experiment. Finally, the target's encoded initial state is in situ correlated with its final state during the second phase accumulation yielding the measurement readout result (compare with our method presented in Fig. 1d). In essence, the coherence time $T_2^{\text{sens}}$ of the sensor grants access even to very weakly coupled targets but grants only limited spectral resolution (i.e., $1/T_2^{\text{sens}}$). The sensor's storage time $T_1^{\text{sens}}$, in contrast, grants identification of targets via an increased spectral resolution (i.e., $1/T_1^{\text{sens}}$) of the undisturbed evolution during the correlation time $T_c$. Thus, the frequency resolution is improved to about 100 Hz in such an in situ correlation spectroscopy measurement[26]. However, as residual phase information is lost during classical storage on $\langle S_z \rangle$, the signal amplitude is decreased

---

**Table 1 Decay times**

| Time | Typical value | Description | Influence |
|---|---|---|---|
| $T_2^*$ | >1 s | Target spin-coherence time | Limits ultimate spectral resolution |
| $T_2^{\text{sens}}$ | $688 \pm 31\,\mu s$ | Sensor spin-coherence time | Limits sensing time (i.e., target access) |
| $T_1^{\text{sens}}$ | $6.3 \pm 0.6\,\text{ms}$ | Sensor spin longitudinal-relaxation time | Affects target and memory coherence |
| $T_2^{\text{mem}}$ | $8.6 \pm 1.3\,\text{ms}$ | Memory spin-coherence time | Limits quantum storage time |
| $T_1^{\text{mem}}$ | $260 \pm 20\,\text{s}$ | Memory spin longitudinal-relaxation time | Limits classical storage time |

Summary of relevant decay time constants for target, sensor, and memory spins. We give typical values in our experiments and describe their influence for in situ correlation spectroscopy

---

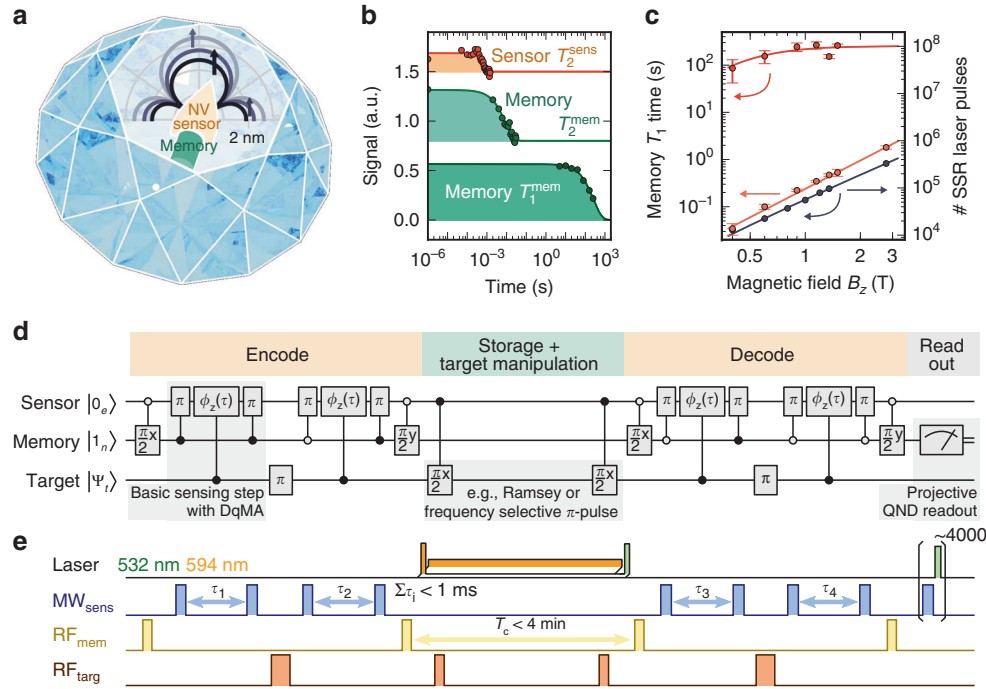

**Fig. 1** The combined sensor–memory spin system. **a** Schematic representation of the sensor–memory pair comprising the electron and $^{14}$N nuclear spin of an NV center in diamond. The inset sketches potential locations of $^{13}$C target spins detected in this work. **b** Longitudinal and transverse relaxation times ($T_1$, $T_2$) of sensor and memory spin (sens, mem). **c** $T_1^{mem}$-scaling with magnetic field strength for three cases: NV center in the negative (red line), the neutral charge state (orange line), and during continuous memory readout (gray line). The shown errors correspond to the standard error of the exponential fit to the decay of the spin state. **d**, **e** Wire diagram and pulse sequence representation of an exemplary Ramsey measurement of target spins, sub-divided into four tasks: encoding; storage, and manipulation; decoding and readout. During encoding and decoding, the sensor phase $\phi$ (dependent on target spin state and duration $\tau_i$) is transferred onto a memory superposition state. This process is efficiently performed by entangling and disentangling sensor and memory with $C_mROT_s$-gates granting the sensor direct quantum memory access (DqMA). Proper conditions of the $C_mROT_s$-gates (i.e., open (closed) circle → sensor $\pi$-flip for memory state "0" ("1")) enable sensitivity to target spins and protection against quasi-static magnetic field noise (see "Methods"). Memory spin $\pi/2$ pulses (phases given by subscripts $x$ and $y$) switch between quantum (i.e., $|0\rangle + e^{i\phi}|1\rangle$) and classical (i.e., $\langle I_z\rangle = \sin\phi$) information storage. Between subsequent basic sensing steps, target spins are flipped, during central storage, this is achieved with high spectral resolution via Ramsey interferometry (i.e., two RF $\pi/2$ pulses). Appropriate Laser illumination enables continuous sensor repolarization into $|0\rangle$, or ionization into the neutral NV$^0$ for decoupling of sensor and target spins during central storage. During decoding, the current target spin state is correlated with its encoded initial state stored as expectation value $\langle I_z\rangle$ on the memory. Finally, the memory state is readout in a single shot[32]

by a factor of $\frac{1}{2}$ on average[9]. Note that under certain conditions also non-in situ correlation spectroscopy is feasible with NV centers in diamond[21].

Apart from measurement back-action, transverse relaxation time $T_2^*$ limits a spin's free precession, and therefore its intrinsic linewidth. However, as the relaxation times of our sensor $T_1^{sens}$ and $T_2^{sens}$ are typically shorter than that of the detected target spins, utilizing a longer living memory is vital to reach the limit imposed by the nuclear spin's inherent $T_2^*$ time.

It has recently been shown, that for these kind of correlation measurements, the nitrogen nuclear spin, present in every NV center, can be used as a quantum memory[9, 39, 40]. Its strong hyperfine coupling to the NV sensor spin ($^{15}$N: 3.03 MHz, $^{14}$N: −2.16 MHz) grants fast memory access and therefore does not noticeably shorten the available phase-accumulation time $T_2^{sens}$. In addition, owing to the quantum nature of the spin, the full quantum information can be stored, resulting in a higher signal contrast. However, the memory's transverse relaxation time $T_2^{mem}$ is also limited by the longitudinal relaxation of the sensor, resulting in quantum storage times on the order of $T_1^{sens}$ (Fig. 1b). The nitrogen memory spin can be efficiently readout in a single-shot measurement with high fidelity simultaneously initializing the memory for the next measurement run[32].

Apart from the intrinsic nitrogen spin, it has also been shown that a particular, few kilohertz-coupled $^{13}$C target spin can be used as a memory that can keep quantum information on the

order of seconds under high-intensity laser illumination[21]. That particular spin was characterized with an unprecedented spectral resolution of ≈0.6 Hz. However, memory access was quite slow and therefore first, uses up almost the total phase-accumulation time $T_2^{sens}$ and second, slows down the readout process considerably (about 100 times in our experimental setting due to the smaller coupling constant). Third, optical illumination ionizes the sensor spin and therefore reduces correlations of subsequent measurements. Furthermore, utilizing a particular $^{13}$C spin does not yield a universal method, and the applied high-intensity laser prevents utilizing the nitrogen spin.

**$^{14}$N nuclear spin as nonvolatile classical memory**. In this work, we present a universally applicable improvement of NV-based NMR correlation spectroscopy at room temperature. We significantly extend the previously reported correlation times and therefore enable spectral resolution beyond the $T_1^{sens}$ limit. To this end, we utilize the expectation value $\langle I_z\rangle$ of the nitrogen nuclear spin as an intermediate classical memory. We achieve robustness of this memory under storage, sensor initialization, readout, and decoupling operations (as opposed to previous work[26]). In particular, we show minutes-long storage times at room temperature, which is far beyond any other measurement timescale in this work (Fig. 1b). Therefore, the $^{14}$N spin can be regarded as a nonvolatile memory. In addition, we develop and compare compatible

decoupling techniques for the target spins to finally demonstrate high-resolution NMR spectroscopy on individual $^{13}$C target nuclear spins within an isotopically purified diamond crystal ([$^{12}$C] = 0.99995). The low $^{13}$C content allows for individual access to very weakly coupled spins and long coherence times $T_2^{\text{sens}}$. These dissipative-decoupling techniques involve either orange (594 nm) laser illumination of the NV center or switching its charge state. Other decoupling techniques like continuously driving the electron-spin transitions are conceivable. Furthermore, we tailor and analyze the filter functions of the detection schemes.

For efficient encoding and decoding of metrology data on the memory, the sensor has direct quantum memory access (DqMA) via mutual entanglement[9] avoiding long-lasting SWAP-gates (Fig. 1d, e). To this end, a superposition state of the memory is entangled with the sensor by use of a $C_m\text{ROT}_s$ gate (i.e., rotation of sensor spin conditional on memory spin state), causing the sensor–memory system to acquire a quantum phase dependent on an external magnetic field (e.g., the Overhauser field of a target spin). A second $C_m\text{ROT}_s$ gate disentangles the two spins, concluding a single sensing step and leaving the acquired phase $\phi$ on the memory superposition state $|0\rangle + e^{i\phi}|1\rangle$ and the sensor in a certain spin projection (e.g., $|0\rangle$ or $|1\rangle$, Fig. 1d). By carefully choosing the control conditions of the $C_m\text{ROT}_s$-gates, the sign of the phase $\phi$ as well as the resulting sensor state can be adjusted. This enables composing dynamical decoupling sequences and influencing free evolution during $T_c$ (see Fig. 1d and "Methods"). Two sensing steps with intermediate short-quantum storage and $\pi$-flip of the target spins comprise one encoding or decoding part (see Fig. 1d, compare ref. [9]). During the central correlation time $T_c$, information is stored as classical expectation value $\langle I_z \rangle = 1/2 \sin\phi$ by applying a $\pi/2$-flip to the memory spin. It is this storage interval $T_c$, which we seek to extend and where we manipulate target nuclear spins with radiofrequency (RF) pulses without influencing the information on the memory. Target spins that are flipped during $T_c$ (and during short-quantum storage throughout encoding and decoding) contribute maximally to the correlation readout result, whereas the effect of quasi-static magnetic field noise is filtered out (see "Methods"). In general, arbitrary NMR pulse sequences can be applied to induce flips of target nuclear spins during the central RF manipulation period. In this work, we either perform an ordinary, frequency selective $\pi$-pulse, or a Ramsey sequence.

Figure 2a and b shows exemplary NMR spectra of $^{13}$C target spins, weakly coupled to NV center ($A$), obtained with our measurement sequence as follows. We access an individual spin efficiently, if the total phase-accumulation time $\tau$ ($\tau = \sum_{i=1}^{4} \tau_i$ in Fig. 1e) is equal to $A_{zz}^{-1}/2$ [9], where $A_{zz}$ is the hyperfine coupling between sensor and target spin ($\tau = 100$ and $200\,\mu s$ in Fig. 2a, b, respectively). The inset in Fig. 2a and b shows a simplified illustration of the measurement scheme comprises coding and storage intervals (compare Fig. 1 and "Methods"). During the whole storage and correlation interval $T_c$, we apply a constant-frequency $\pi$-pulse to the target spins (430 and 860 $\mu s$ in Fig. 2a, b), which has a narrow spectral range of about $\pm 1$ and $\pm 0.5$ kHz. We acquire a spectrum by stepping the $\pi$-pulse frequency. Although the upper spectrum in Fig. 2a reveals one resonance at the bare $^{13}$C Larmor frequency, the lower spectrum and its zoom-in in Fig. 2b shows two target spins, $A_1$ and $A_2$, shifted by their individual hyperfine coupling $A_{zz}$. To acquire the spectrum, we have set the sensor spin into its non-magnetic $|m_S = 0\rangle$ or magnetic $|m_S = 1\rangle$ state during $T_c$, respectively. The latter case switches on coupling and therefore the possibility to manipulate individual target spins conditional on their coupling strength. Note that this option becomes available because we do not use the sensor spin for information storage as in earlier experiments[26]. Possible locations of the two spectrally resolved $^{13}$C spins with respect to the NV center in the diamond lattice are displayed in Fig. 2c.

Following the results of Fig. 2b, we set the RF $\pi$-pulse frequency resonant to target spin $A_1$, with a duration of 860 $\mu s$. When changing the total phase-accumulation time $\tau$, the measurement signal oscillates with the coupling strength (Fig. 2d), characteristic for a single target spin. We establish maximum correlation of sensor and target, and therefore prepare a strong projective measurement of the target, by setting $\tau = 300\,\mu s$ [30].

With these settings, we perform a Ramsey experiment selectively on target spin $A_1$ (Fig. 1b). Figure 2e shows the resulting Ramsey oscillation, which decays exponentially with a time constant of 4.5 ms, corresponding to a linewidth of 71 Hz. In contrast, the expected decay constant of a single $^{13}$C spin in such an isotopically purified diamond is <1 Hz (the coupling of two $^{13}$C spins at average distance is around 0.07 Hz). The main reason for the comparably fast signal decay are the probabilistic sensor spin flips on a timescale $T_1^{\text{sens}}$ leading to fluctuating magnetic fields and hence resonance frequency shifts for both, memory and target spin (Eq. (1)).

**Dissipative decoupling of memory and target spins from the sensor spin.** As we cannot decrease the sensor's dissipation rate $1/T_1^{\text{sens}}$ at room temperature[27], we need to mitigate its effect on the stored metrology information and on the target spins. The sensor's dissipation affects the memory by dephasing the stored information but not $\langle I_z \rangle$. Therefore, the memory spin's expectation value $\langle I_z \rangle$ is indeed a natural choice for information storage. When measuring the $T_1^{\text{mem}}$ time of the memory spin for magnetic fields in the range of 0.6 to 1.5 T, we achieve values above 100 s, which increase up to $260 \pm 20$ s for magnetic fields above 1 T (see Fig. 1b, c and "Methods"). Lifetimes that long are due to the low-noise-spectral density at the memory's resonance frequency (e.g., no bath of other nitrogen nuclear spins) and due to efficient decoupling from the sensor spin[32]. Apart from long memory lifetimes under dark conditions, during continuous memory readout information decays after about 2 s at 3 T, which equals about 200 memory readouts and about $10^6$ sensor initialization steps (Fig. 1c–e).

In addition to a protected memory, we need to prevent decoherence of the target spins. This can be achieved for example by fast coherent (e.g., Rabi oscillations) or incoherent (i.e., dissipation) flips of the sensor spin. Here, we chose a dynamic decoupling approach, either via continuous optical pumping into the $|0\rangle$ eigenstate, or by exploiting the increased electron-spin dissipation rate in the NV center's neutral charge state (Fig. 1e)[41]. Analogous to motional averaging in liquid-state NMR experiments, the effect of the coupling between sensor and target spins is averaged out and the coherence time of the target spins is prolonged (compare refs. [21, 42, 43]). Therefore, we seek to increase the sensor-spin flip rate $\Gamma$ much beyond the sensor–target coupling strength $\Gamma \gg A_{zz}$ for the target spin linewidth $\delta\nu = \left(\pi T_2^*\right)^{-1}$ to scale as

$$\delta\nu \propto A_{zz}^2 \Gamma^{-1} \tag{2}$$

(see "Methods").

In the case of continuous optical sensor-spin initialization, the spin flip rate is related to the excitation rate as $\Gamma \propto \gamma_{\text{exc}}$. Hence, a higher excitation rate is beneficial for the decoupling effect on the target nuclear spins[21]. In addition, the sensor-spin initialization probability into $|0\rangle$ of $\approx 98\%$ [36] reduces the average coupling $A_{zz}$ and hence, the pre-factor in Eq. (2). However, high excitation rates also cause decay of the memory qubit spin-expectation value $\langle I_z \rangle$, both by ionization of the NV center to its neutral charge state with rate $\gamma_{\text{ion}} \propto \gamma_{\text{exc}}^2$ and by higher occupation probability of the electronic excited state[32, 37, 44–46]. The long storage lifetime during memory readout (Fig. 1c) suggests that the dissipation effect on the memory due to optical excitation is minor. Ionization into NV$^0$ prevents access to the sensor and memory spin, is more likely

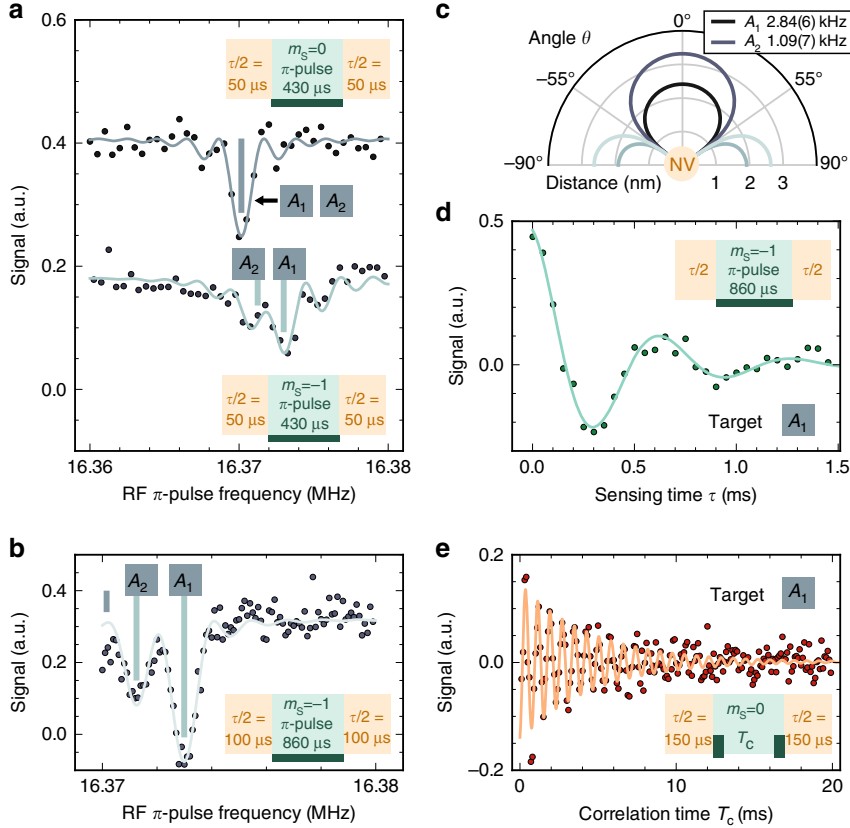

**Fig. 2** Spectroscopy of distant, weakly coupled $^{13}$C spins limited by sensor dissipation. **a** Spectra of the $^{13}$C target spins surrounding the NV center. For the upper (lower) $^{13}$C spectrum, the sensor spin is in its non-magnetic $|m_S = 0\rangle$ (magnetic $|-1\rangle$) state during the storage interval. Hence, the upper curve shows the bare Larmor frequency common to all $^{13}$C nuclear spins in the interaction range, whereas the lower curve shows additional hyperfine coupling offsets $A_{zz}$ for the two distinct nuclei $A_1$ and $A_2$. The total phase-accumulation time is set to $\tau = 100\,\mu s$ (orange parts of insets) and the RF $\pi$-pulse (dark green part of insets), which flips the $^{13}$C spins during the storage interval, is set to $T_c = 430\,\mu s$ (light green part of insets). **b** Increased sensing and storage time ($\tau = 200\,\mu s$ and $T_c = 860\,\mu s$, respectively) generate a close-up spectrum of target spins $A_1$ and $A_2$. Resonance frequencies are marked with vertical lines. **c** Possible angles $\theta$ and distances $d$ of the two $^{13}$C spins relative to the NV center. The functions are $d \propto \left(A_{zz}^{-1}\left|3\cos^2\theta - 1\right|\right)^{1/3}$. For the identified hyperfine interactions $A_{zz}$, $3\cos^2\theta - 1$ is positive (dark line parts for $\theta = -54.7°...54.7°$). **d** Variation of the overall phase-accumulation time $\tau$ for an RF $\pi$-pulse selective on the strongest coupled $^{13}$C spin (i.e., $A_1$, $\nu_{RF} \approx 16.373$ MHz, $T_c = 860\,\mu s$) reveals an oscillation, which confirms the coupling strength of 2.8 kHz. **e** For a phase-accumulation time of $\tau = 300\,\mu s$ the memory state is almost maximally correlated with the flip of target $A_1$. During storage, we perform a Ramsey oscillation on the $^{13}$C spin (inset shows two dark green $\pi/2$ pulses), which is converted into a memory signal and decays with the memory lifetime, limited by dissipation due to the sensor

in this decoupling scenario and should therefore be avoided. By carefully choosing the excitation power at a wavelength of 594 nm[37, 46], one can set the ionization limit on the linewidth $\delta\nu$ equal to the dissipative-decoupling limit according to Eq. (2) (see "Methods"). We measure the Ramsey oscillation decay of target spin $A_1$ for varying optical excitation power and reveal an enhancement of the target spin-coherence time by almost a factor of four up to $T_2^* = 17.4 \pm 4.2$ ms, at a repumping laser power of 6.3 µW (Fig. 3b). The corresponding Fourier transformation reveals a sharp peak with a FWHM of $18.3 \pm 4.3$ Hz.

Further increasing the excitation rate causes the NV center to ionize faster, resulting in a faster loss of signal and an apparently shorter $T_2^*$ of the target spin (Fig. 3a). However, the latter loss of signal neither implies decay of the classical information stored on the memory nor faster intrinsic dephasing of the target spin. It is due to the lack of memory access. To restore access, we recover the negative charge state (i.e., NV$^-$) after the correlation time $T_c$ and before the final decoding part of the sequence by a short green (532 nm) laser pulse (Figs. 1d, e and 4c). It turns out, that the uncharged state of the NV center can also be used for decoupling of target spins, which constitutes our second decoupling approach. Owing to the intrinsic fast spin flip rate $\Gamma_{NV}^0$ of the electron spin in NV$^0$ [36], no

laser excitation during $T_c$ is required. Instead, these fast flips lead to an effective decoupling for small couplings $A_{zz} \ll \Gamma_{NV}^0$.

We benchmark the two decoupling techniques on a different NV center (B), with a $^{13}$C target spin $B_1$ more weakly coupled than target $A_1$ (i.e., $-1.8$ kHz instead of 2.8 kHz, see Fig. 4a–c). When utilizing the continuous optical repolarization method, we obtain a Ramsey oscillation with a decay constant of $23.8 \pm 2.9$ ms and the Fourier transform reveals a corresponding linewidth of $13.3 \pm 1.6$ Hz (Fig. 4b). For comparison, we measure the linewidth with the NV center being in its neutral charge state (Fig. 4c, d). To this end, we switch the NV center during the central correlation interval first into the neutral charge state by a strong orange laser pulse (100 µW, 1 ms duration), then perform a Ramsey experiment, and finally recover the negative charge state by a green laser pulse prior to the decoding interval (Fig. 4e), resulting in a linewidth of 40.8 Hz. Note, that the restoration fidelity of NV$^-$ is limited to about 70%[36], which directly translates to a reduced signal intensity.

For the purpose of protecting the coherence of target spin $B_1$, optical repolarization is clearly superior to ionization into NV$^0$. However, when considering the scaling of the expected target spin-coherence lifetime with decreasing coupling for both

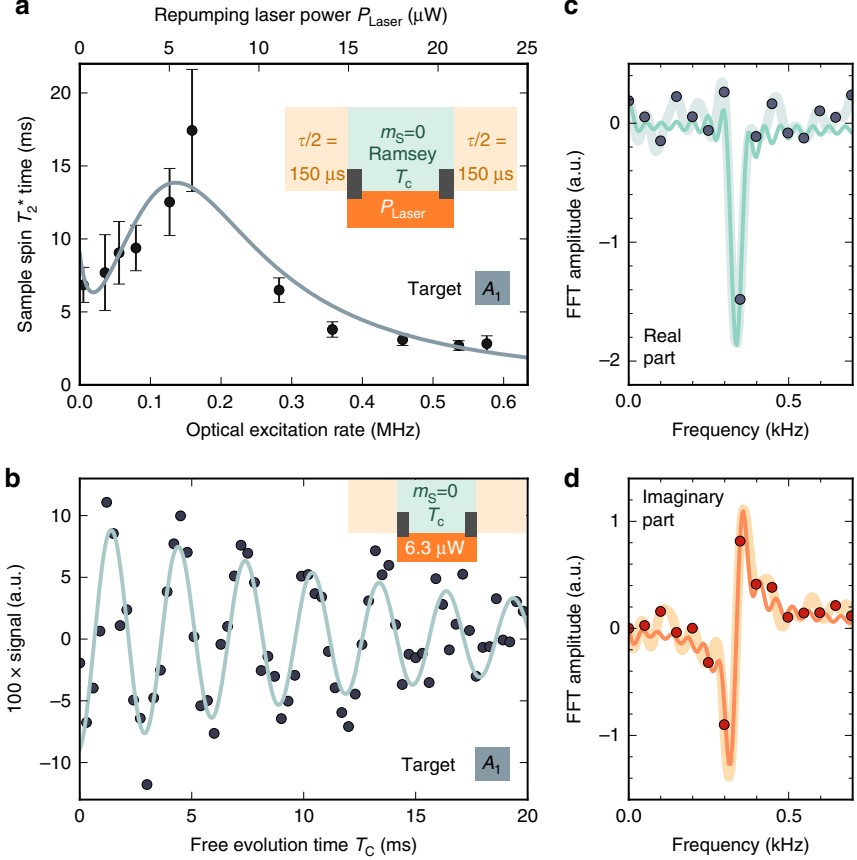

**Fig. 3** Dissipative decoupling of target spins during classical storage on quantum memory. **a** During storage (green background) the sensor-spin flips stochastically into a magnetic state $|\pm 1\rangle$ and an orange repumping Laser (594 nm) excites the NV and repolarizes the sensor spin predominantly into the non-magnetic state $|0\rangle$. We perform Ramsey measurements on target spin $A_1$ and record the $T_2^*$ time depending on the applied optical excitation rate. The shown errors correspond to the standard error of the exponential fit to the decay of the spin state. We fitted simulated coherence lifetimes to the measured results using three parameters, (i) the conversion from applied laser power to excitation rate, (ii) the ionization probability, and (iii) the probability for the $|\pm 1\rangle$ excited state to decay via the metastable state into the $|0\rangle$ ground state (see "Methods"). Three distinct regions are observed. The first decrease is induced by a slow excitation and depolarization into the $|\pm 1\rangle$ spin state. As soon as the excitation rate exceeds the coupling strength, the coherence time starts to increase. At a certain excitation rate, the sensor is ionized before the measurement can be concluded, resulting in loss of signal. **b** Longest living Ramsey oscillation of target $A_1$ for an applied laser power of 6.3 μW. The solid line is a fit of a decaying cosine oscillation, with a decay time $T_2^* = 17.4 \pm 4.2$ ms. **c**, **d** Real (imaginary) part of the Fourier transformed signal of (**b**). Although the dark gray (orange) dots represent the raw transform, the bold, light gray (orange) line represents the transformation of the zero-filled input signal. The thin, green (orange) line shows a fit, where the real and imaginary parts of the Fourier transform are fitted simultaneously resulting in a linewidth of $18.3 \pm 4.3$ Hz

methods, we observe crossing at a coupling strength of $A_{zz} \approx 400$ Hz and a sensor-affected target spin-coherence time of $T_2^* \approx 150$ ms (Fig. 4d). To this end, we extrapolate the performance of or decoupling techniques for smaller couplings $A_{zz}$ as follows. Comparing our $NV^0$ results with those of ref. [36] (i.e., $A_{zz} = -1.8$ kHz and $T_2^* \approx 8$ ms vs. $|A_{zz}| = 6.06$ MHz and $T_2^* \approx 6$ μs), we conclude that in the present case, we are well within the motional averaging regime and therefore the coherence time is expected to scale as $T_2^* \propto A_{zz}^{-2}\Gamma_{NV^0}$.

For the optical repolarization technique, however, we have to set an optimum excitation rate, which decreases with decreasing coupling strength as $\Gamma_{exc} \propto A_{zz}^{2/3}$ (see "Methods"). Hence, we expect a power-law scaling of $T_2^* \propto A_{zz}^{-4/3}$. Given our experimental data, we simulate the expected $T_2^*$ for target spins with different couplings (see "Methods") and get a curve shown in Fig. 4d, which agrees with the expected power-law scaling.

For comparison, we consider an NV center remaining in the negative charge state during the correlation time without any dissipative-decoupling technique. Then, down to sensor–target coupling strengths of about 100 Hz the target spin-coherence time $T_2^*$ would be pinned at 9 ms and only for even smaller couplings

$T_2^*$ would rise. For example, a target spin-coherence time of 18 ms is expected for a coupling strength of $A_{zz} \approx 35$ Hz (see diamond in Fig. 4d). Such a coupling is found for a $^{13}$C–NV distance of up to 10 nm or a $^1$H–NV distance of up to 17 nm (Fig. 4e).

## Discussion

In this work, we have implemented a hybrid sensor system, which comprises an electron-spin sensor for magnetic fields and a nuclear spin memory capable of quantum and classical metrology data storage. Although quantum storage was used for efficient encoding and decoding of the target spin states as in earlier experiments[9], classical storage on the memory's $\langle I_z \rangle$ expectation value proved to be very robust and exhibited storage times on the order of seconds to minutes for various conditions. In particular, such lifetimes are achieved while reading out the memory, reinitializing the sensor, and decoupling the target from the sensor. As these lifetimes are beyond all applied target evolution times and beyond typical NMR measurement times, our sensor can be regarded as nonvolatile. To exploit such hybrid sensor system, we have adapted universally applicable in situ correlation

spectroscopy. An experimental demonstration of a similar spectroscopy concept appeared as preprint along with ours[47].

The demonstrated universal sensor–memory approach enabled performing high-resolution spectroscopy on several target $^{13}C$ spins in diamond at room temperature. Here, the long-lived intermediate classical storage should have increased the spectral resolution far beyond the capabilities of the sensor alone. However, we encountered that spectral resolution of target spins stayed pinned due to sensor relaxation. We have developed dissipative-decoupling techniques that on the one hand preserve the information stored on the memory and on the other hand efficiently reduce the deleterious effect of the sensor on the target spins. Consequently, we are able to measure nuclear magnetic resonance linewidths of single $^{13}C$ spins in diamond of 13.3 Hz.

Although we have confirmed decoupling performance for target spins with coupling strengths down to a few kHz, we have also simulated the performance for spins up to the coherent coupling limit of around 300 Hz and beyond. We set the latter limit tentatively by the maximum coherence time of the NV sensor spin of up to 3 ms at room temperature measured so far[48]. For such target spins, dissipative decoupling is expected to improve the coherence time and hence spectral resolution by a factor of about 20.

Below a coupling strength of 300 Hz down to about 30 Hz, we enter the weak measurement regime, where the encoded and decoded sensor signal of individual target spins decreases along with a reduced measurement back-action onto the target spins[30]. However, the dissipative back-action would still deteriorate target spins. For example, in nanoscale NMR experiments[15, 16, 19, 20], where clusters of proton spins at a distance of ~10 nm to the NV sensor and a corresponding coupling of about 160 Hz are detected, the dissipative back-action of the sensor on the target

spins still limits their coherence times to about 10 ms. Hence, in several previous NV-based NMR measurements on external nuclear spins[15, 16, 19, 20], the NV sensor itself has limited the NMR linewidth. Here, our dissipative-decoupling methods would have increased the target lifetime by a factor of about 100, i.e., beyond one second. Such target coherence lifetimes would allow linewidths encountered in conventional NMR. Therefore, our demonstrated sensing, storing and decoupling approach enables sensor-unlimited spectroscopy. One prime application would then be nanoscale NMR spectroscopy with the capability to resolve J-couplings and chemical shifts of a few molecules or proteins. So far the spectral resolution in such experiments, however, was not only limited by the sensor but also by too short sensor–target interaction times or dipolar broadening.

All our decoupling methods needed either continuous mild or short and intense laser illumination. For some target systems that might already affect their state (e.g., optically active spin systems). As a solution the NV center can be switched into the neutral charge state by purely electrical means[49]. Recently, another charge state of NV centers has been identified, namely NV$^+$[50]. It turned out to have an electron-spin angular momentum $S = 0$. Therefore, its nuclear spin preserves quantum and classical information. Hence, it promises to be the ideal storage charge state. In such a scenario, there might not be any limiting effect of the NV center on coherence times of target spins.

For even weaker coupled spins, it becomes hardly possible to detect single target spins and we start entering the regime of a classical sample, where both measurement and dissipative back-action become less important. In this classical regime, in situ correlation measurement techniques as described here, which avoid probing and therefore disturbing the targets during the correlation period, are not essential. The target can rather be

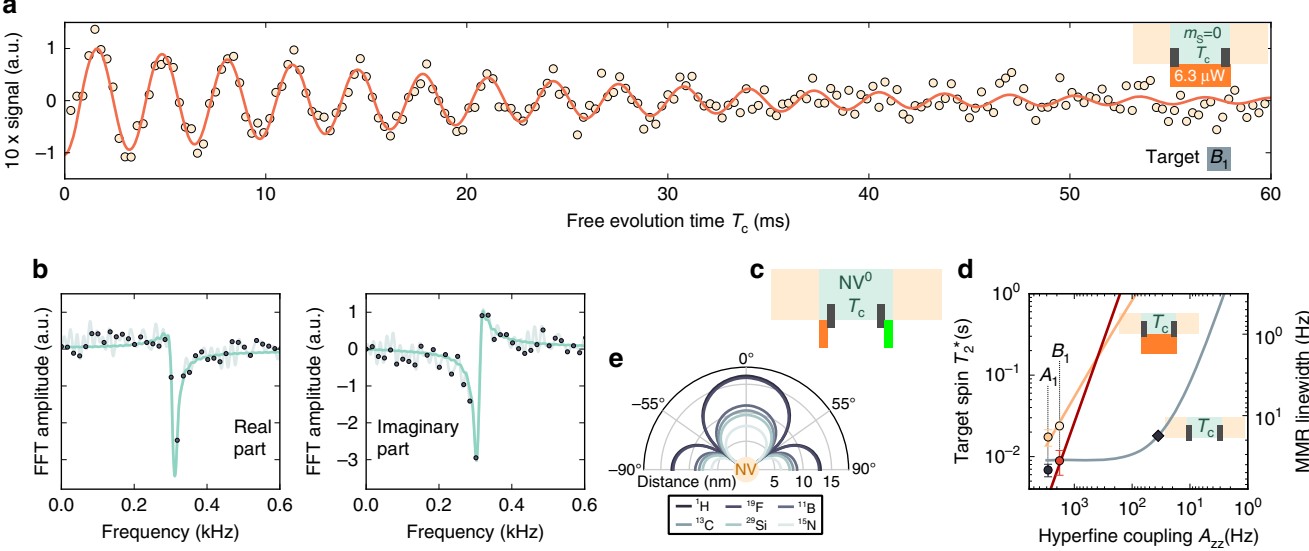

**Fig. 4** Scaling of decoupling techniques for improved spectral resolution. **a** Ramsey oscillations of a different NV with a −1.8 kHz-coupled $^{13}C$ spin (target $B_1$) during constant sensor-spin repumping. The decay constant of the oscillation is 23.8 ± 2.9 ms. **b** Fourier transform of (**a**). Analogous to Fig. 3c and d, the dark gray points are the transform of the unprocessed measurement data, the light gray line is the transform of the zero-filled signal. The green line is the simultaneous fit of the real and imaginary part of the transformation. The linewidth was determined to be 13.3 ± 1.6 Hz. **c** Simplified measurement scheme of the target decoherence, whereas the sensor is in its neutral charge state. Before applying the first $\pi/2$ pulse on the target spin, the NV center is pumped to the NV$^0$ state by applying a 1 ms orange laser pulse with a power of 100 μW. The charge state is afterwards recovered by applying a short green laser pulse. **d** Simulated sensor limits on coherence times (lines) of a target nuclear spin coupled to an NV center, when using one of the aforementioned decoupling methods. The gray curve is for the plain NV$^−$, the red for the NV$^0$ case (see **c**). The orange line shows the coherence time limit, when using the optimal optical repolarization rate for every coupling (see "Methods"). All measured coherence times of this work are displayed as color-coded circles. The shown errors correspond to the standard error of the exponential fit to the decay of the spin state. The gray diamond marks the hypothetical coupling strength at which the plain NV$^−$ exerts a dissipative-decoupling effect, which doubles the target spins coherence time limit. In **e**, the possible location of nuclear spins with said coupling strength of different spin species are shown (see diamond in **d**)

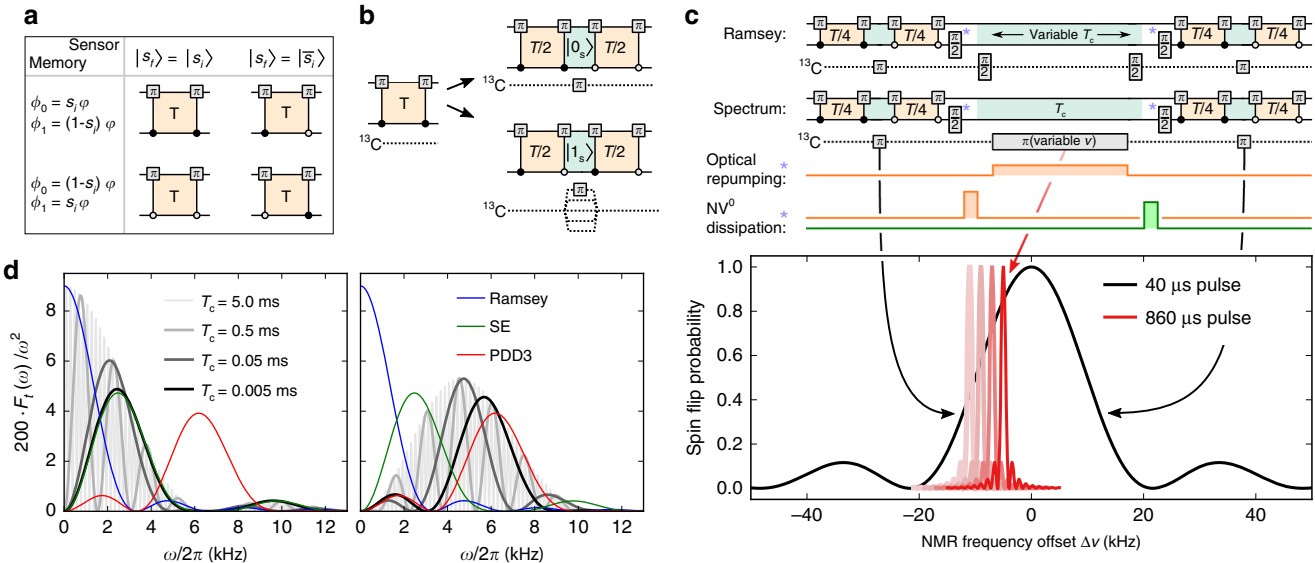

**Fig. 5** Design of sensing sequences. **a** Four sensing steps with DqMA used to create different sensing tasks. Every sensing step starts with the sensor spin prepared in an eigenstate and the memory spin in a superposition state. These sensing steps have two tasks: accumulate a phase on one of the components of the memory spin (i.e., $\phi_0$ or $\phi_1$), and either flip or keep the sensor-spin state (i.e., $|s_f\rangle = |s_i \oplus 1\rangle$ or $|s_f\rangle = |s_i\rangle$). **b** Refinement of a single Ramsey-type sensing step into an echo-type sequence comprising two sensing steps and an intermediate flip of target spins to facilitate assignment. Flips can be selective, by turning on the hyperfine gradient during the pulse (lower part). **c** Sensing sequences used in Figs. 2–4. Optional elements are marked with a blue star. The optional $\pi/2$ pulses on the memory spin convert quantum into classical storage and vice versa. The optional laser pulse sequences realize different decoupling methods. The outer short $\pi$ pulses on the $^{13}$C spins filter the accumulated signal in a broad spectral range of NMR frequencies, whereas the central, longer-lasting $\pi$ pulse provides high-frequency resolution. The lower part illustration the spectral filters for $^{13}$C target spins from the above sequence. The overall signal results from target spins that are flipped by all three pulses. **d** Noise filter functions of the used detection sequences acting on sensor and memory, in comparison to commonly known detection schemes acting only on the sensor (spin echo (SE), periodic dynamical decoupling (PDD)). On the left, the sensing steps as shown in (**a**) are used for encoding and decoding, whereas the right graph incorporates the echo-type sequence, shown in (**b**)

measured continuously without any back-action and a correlation can later-on be performed on the available measurement data as, for example, in fluorescence correlation spectroscopy[51].

## Methods

**Experimental setup and diamond sample**. Our experiment consists of a home-built confocal microscope with 532 and 594 nm excitation lasers, referred to as green and orange lasers, respectively. The cw lasers can be switched on and off on the timescale of ~10 ns with an acousto-optical modulator (AOM) and reaches on/off intensity ratios of up to $10^6$. To completely block illumination lasers, we can additionally flip a beam block into the optical path on timescales of one second. The emitted photoluminescence of single NV centers is collected via an oil-immersion objective with an NA of 1.35 and detected by a single photon-counting detector (avalanche photodiode, APD). The diamond crystal is positioned inside a $B_z = 3$ T, $B_{x,y} = 0.2$ T superconducting vector magnet, with the diamond's surface normal pointing along the main magnetic field axis[52]. The magnet is mainly operated at a field of about 1.5 T, which shifts the NV sensor-spin resonance frequency to about 40 GHz. Spin resonance is detected optically (ODMR) via spin state-dependent fluorescence of single NV centers[53]. Microwave (MW) radiation for hyperfine selective sensor-spin manipulation is generated by amplitude mod-ulation (at frequencies of ~1 GHz) of a carrier signal (at $\approx$40 GHz), utilizing a double-balanced harmonic mixer and an arbitrary waveform generator (AWG). The microwaves are guided through coaxial cables and a coplanar waveguide. The latter is tapered to around 100 $\mu$m at the location of the NV for a larger MW field strength. The RF signal used to manipulate the nuclear spins is created directly by the AWG and guided through a 20 $\mu$m copper wire spanned across the diamond perpendicular to the CPW direction.

The diamond crystal is a polished, (111)-oriented slice (2 mm × 2mm × 88 $\mu$m) from a high pressure and high temperature (HPHT) diamond crystal (5.3 mm × 4.7 mm × 2.6 mm). The original diamond is isotopically enriched with a $^{12}$C-concentration of 99.995%. The crystal was irradiated by 2 MeV electrons at room temperature to the total fluence of $1.3 \times 10^{11}$ cm$^{-2}$ and annealed at 1000 °C (for 2 h in vacuum) to create bulk NV centers from intrinsic nitrogen impurities.

The isotopical enrichment of the diamond enables detection of individual $^{13}$C spins with hyperfine coupling strengths of a few kHz, in the first place. Otherwise, the spectral density of $^{13}$C spins in a range of a few kHz around the bare Larmor

frequency would hamper the discrimination of single spins. Furthermore, the average sensor coherence time increases with decreasing $^{13}$C concentration[23].

**Detailed measurement sequences**. In the following, we explain the applied measurement sequences in more detail. Therefore, we take a closer look on the individual sensing parts to reveal how magnetic field sensing information is acquired and stored on the memory and how it is spectrally filtered to be sensitive mainly to the target spins.

A full measurement sequence consists of two sensing (encoding and decoding) and one intermediate storage interval. In the experiments performed here, the encoding as well as the decoding interval again are each sub-divided into two sensing steps and one intermediate, much shorter, storage step (Fig. 5b, c). A proper timing of sensing and storing constitutes a dynamical decoupling sequence that renders the sensor–memory system sensitive to target spins and almost insensitive to other "noise" sources.

During sensing steps, the sensor–memory system is sensitive to any magnetic field source and thus picks up a corresponding phase $\phi$. During storage, the latter sensitivity is effectively switched off and changes to the environment (e.g., the target spins) can be performed without influencing the intermediate sensing result on the memory.

We utilize entangled states of the sensor and memory qubits for sensing and simultaneous DqMA during sensing steps (e.g., $|\Psi_{s,m}\rangle = e^{i\phi_0}|00\rangle + e^{i\phi_1}|11\rangle$). The initial qubit states $|0\rangle$ and $|1\rangle$ of sensor and memory are related to the magnetic quantum numbers $m_S = 0, -1$ and $m_I = 0, +1$ for the NV electron and nuclear spin, respectively. While, the sensor spin is polarized by laser illumination, the memory spin state is reported by the preceding readout step. Here, we use this information for postselection of statistical initialization. One might also use deterministic memory initialization as demonstrated in ref. [14]. Initial and final sensor–memory quantum states, as well as states during the storage steps are product states $|\Psi_s\rangle \otimes |\Psi_m\rangle$ of a sensor eigenstate $|\Psi_s\rangle = |0\rangle, |1\rangle$ and a memory superposition state $|\Psi_m\rangle$,

$$
\begin{aligned}
|\Psi_s\rangle \otimes |\Psi_m\rangle &= |\Psi_s\rangle \otimes \left(e^{i\phi_0}|0\rangle + e^{i\phi_1}|1\rangle\right) \\
&= e^{i\Sigma\phi/2}|\Psi_s\rangle \otimes \left(e^{i\Delta\phi/2}|0\rangle + e^{-i\Delta\phi/2}|1\rangle\right),
\end{aligned}
\tag{3}
$$

where the phases $\phi_i$ contain the stored magnetic field information (Figs. 1b and 5). The initial phases are $\phi_{i=0,1} = 0$. Note that $\Sigma\phi = \phi_0 + \phi_1$ constitutes a global phase and only the interference term $\Delta\phi = \phi_0 - \phi_1$ is accessible at the final read out.

During sensing steps, NOT-gates on the sensor spin conditional on the state of the memory spin (i.e., $C_mROT_s$-gates) entangle and disentangle sensor and memory (Fig. 5a). Although sensor and memory are entangled, a phase $\varphi$ is accumulated, which linearly depends on the phase-accumulation time $\tau$ and the local magnetic field, for example, due to hyperfine-coupled target spins. The actual conditions of the $C_mROT_s$-gates, their order, and the initial sensor state determine to which storage phase $\phi_i$, the sensing phase $\varphi$ is added. During all three storage steps between the sensing steps, we manipulate target nuclear spins with RF pulses to induce a measurement signal. Target spins that are flipped in all three storage periods contribute maximally to the accumulated signal, whereas the total phase due to quasi-static magnetic field noise is filtered out (i.e., does not contribute to $\Delta\phi$).

The last memory spin $\pi/2$ pulse during decoding converts the final memory phase into a population difference of states $|0\rangle$ and $|1\rangle$. This population is then read out in a projective non-demolition manner[30, 32] yielding a single bit of information. Averaging then yields a probability with visibility of up to about 0.65 in our case due to limited readout fidelity and charge state switching[36], decay mechanisms decrease this value further. All signals displayed in this work are referenced values where the probabilities are subtracted from those with a $\pi$-phase-shifted last memory spin $\pi/2$ pulse. Accordingly, the maximum signal contrast or visibility can be as large as 1.3 in our case.

Figure 2a and b shows exemplary NMR spectra of weakly coupled $^{13}C$ target spins obtained with our measurement sequence as follows (Fig. 5c). We set the total phase-accumulation time to $\tau = 100$ and $200\,\mu s$ in Fig. 2a and b, respectively. During first and last storage time, we perform each one $\approx 40\,\mu s$-long RF $\pi$-pulse on all target spins within a spectral range of about $\Delta\nu = \pm 10\,kHz$ around the RF frequency (Fig. 5c). Their main purpose here is to enable sensitivity to all the target spins despite the dynamical decoupling sequence of the sensor spin. During the correlation time $T_c$, however, we apply a long $\pi$-pulse (430 and 860 $\mu s$ in Fig. 2a, b), which in turn is selective in a narrow spectral range of about $\pm 1$ and $\pm 0.5\,kHz$ and therefore determines spectral resolution (Fig. 5c). Only those target spins contribute maximally to the signal, which are flipped during all three storage steps.

Although the upper spectrum in Fig. 2a reveals one resonance at the bare $^{13}C$ Larmor frequency, the lower spectrum and its zoom-in in Fig. 2b show multiple target spins shifted by their individual coupling $A_{zz}$. To this end, we have adapted the condition of the $C_mROT_s$-gates during the sensing steps, such that the sensor is in its non-magnetic $|0\rangle$ or magnetic $|1\rangle$ state during $T_c$ (Fig. 5a, b). The latter case switches on coupling and therefore the possibility to manipulate individual target spins conditional on their coupling.

The sensing steps as used in this work, expect the sensor–memory state to be of the form like in Eq. (3) with $|\Psi_s\rangle = |s_i\rangle$ prior to the sensing step. The state after the sensing step is of the same form with $|\Psi_s\rangle = |s_f\rangle$ ($s_i, s_f \in \{0, 1\}$). Sensing steps can therefore flip or not flip the sensor state, while adding a phase $\varphi$ to either phase $\phi_0$ or $\phi_1$ of the memory superposition state (Eq. (3)). We end up with four different gates, adding a phase to one of the two memory states, while flipping or not flipping the sensor spin (Fig. 5a). It can be seen, that two identical consecutive $C_mROT_s$-gates separated by an evolution time $\tau$ do not change the sensor-spin state state (i.e., $|s_f\rangle = |s_i\rangle$), however, accumulate a phase onto one or the other memory spin state depending on the required condition of the control qubit (filled circle → conditional on state $|1\rangle$, open circle → conditional on state $|0\rangle$, see Fig. 5a left column). By using two $C_mROT_s$-gates with different conditional states, the sensor state can be flipped in addition to the phase accumulation (i.e., $|s_f\rangle = |\overline{s_i}\rangle$, see Fig. 5a right column). Depending on the initial sensor state $s_i$ and the conditional state of the first $C_mROT_s$-gate of the pair, the phase $\varphi$ adds either to $\phi_0$ or $\phi_1$ (identical for rows of Fig. 5a).

The overall measurement sequences should constitute dynamical decoupling sequences similar to a spin echo (SE), a Carr–Purcell–Meiboom–Gill (CPMG) or a periodic dynamical decoupling (PDD) sequence[54]. Therefore, the phases $\varphi$ accumulated during successive sensing steps should be added to opposite phases $\phi_0$ or $\phi_1$ to mimic the $\pi$-pulse effect of the standard dynamical decoupling sequences, which constantly switches the magnetic field sensitivity between 1 and $-1$. Different from the standard techniques, our sequence of sensing steps is interleaved with storage times exhibiting no magnetic field sensitivity. The magnetic field sensitivities at times $t'$ during the run of a dynamical decoupling sequence of duration $t$ are given by the function $f(t, t') \in \{-1, 0, 1\}$. Given the requirement of switching sensitivities and the option to set a certain sensor eigenstate during individual storage intervals the proper quantum gates for the series of sensing steps can be constructed according to previous paragraph. Switching of sensitivities also requires flips of target spins during each storage period in order to be sensitive to their field exerted on the sensor.

As for standard dynamical decoupling sequences, we can deduce a filter function $F(\omega t)$ of our full sequence, where $\omega$ is the angular frequency of a potential oscillating magnetic field and $t$ is the total duration of a sequence[54]. Then, $F(\omega t)\omega^{-2}$ expresses the spectral sensitivity to magnetic field noise[54, 55]. The decay of coherent phase information $W(t) = e^{-\chi(t)}$ can then be expressed via

$$\chi(t) = \pi^{-1} \int_0^\infty d\omega S(\omega) \frac{F(\omega t)}{\omega^2}, \qquad (4)$$

where $S(\omega)$ is the noise-spectral density. The filter function is obtained by Fourier-transforming the magnetic field sensitivity function $f(t, t')$ of the measurement

sequence with respect to $t'$ yielding $\tilde{f}(t, \omega)$.

$$F(\omega t) = \frac{\omega^2}{2} \left| \tilde{f}(t, \omega) \right|^2 \qquad (5)$$

The filter functions for the measurement sequences performed here and for the slightly simpler version of ref. [9] are

$$F(\omega t) = 32 \sin^2 \frac{\eta_\tau \omega t}{8} \sin^2 \frac{(2 - \eta_\tau - 2\eta_{T_c})\omega t}{8}$$
$$\times \cos^2 \frac{(1 + \eta_{T_c})\omega t}{4} \qquad (6)$$

and

$$F(\omega t) = 8 \sin^2 \frac{\eta_\tau \omega t}{4} \sin^2 \frac{(2 - \eta_\tau)\omega t}{4}, \qquad (7)$$

respectively, where $\eta_\tau$ and $\eta_{T_c}$ are fractional phase accumulation and central correlation times normalized by the total sequence duration $t$ and obeying $0 \leq \eta_\tau + \eta_{T_c}, \eta_\tau, \eta_{T_c} \leq 1$.

Examples of both filter functions multiplied by $2/\omega^2$ are plotted in Fig. 5c for a total phase-accumulation time $\tau = 0.3\,ms$, duration of first and last storage interval each with RF $\pi$-pulse of 0.04 ms (where applicable) and variable central correlation time $T_c$ up to 5 ms. For comparison, the filter functions for Ramsey, SE and PDD sequence, each with an equal total phase-accumulation time, are displayed. On the left panel, the sequence of ref. [9] is shown (i.e., $\frac{\tau}{2} - T_c - \frac{\tau}{2}$) and on the right panel, the current sequence (i.e., $\frac{\tau}{4} - \pi_{RF} - \frac{\tau}{4} - T_c - \frac{\tau}{4} - \pi_{RF} - \frac{\tau}{4}$). Apparently, the first filter function resembles that of a spin echo for small correlation times. However, for large $T_c \gg \tau$ it becomes a fast oscillating function under the envelope of a Ramsey filter function. Thus, it gets sensitive to small frequency noise. The sequence used in the current paper circumvents this issue by adding another filter step into initial and final sensing interval. For negligible storage times, the new sequence resembles a PDD3 sequence and for large $T_c$ it also shows fast oscillations under the envelope of a SE filter function. Hence, sensitivity to small frequency noise is reduced.

**Characterization of sensor and memory spin qubits.** In the NV center hybrid sensor, the electron and nuclear spin have fundamentally different tasks because of their different properties. Although the electron spin is very susceptible to magnetic fields, the nuclear spin is almost unaffected by it. Yet, the nuclear spin is quite strongly coupled to the electron spin when compared to present relaxation rates. Hence, the electron spin serves as the primary transducer from magnetic fields to a quantum phase (sensor), whereas the nuclear spin is ideally suited for storage of the latter phase (memory). Here, we briefly characterize both spins under the current sample and setup conditions. The spin Hamiltonian describing the combined sensor–memory and the target spins is

$$\begin{aligned}
H &= H^{sens} + H^{mem} + H^{tar} + H^{coupl} \\
&= DS_z^2 + \tilde{\gamma}^{sens}B_z S_z + \tilde{\gamma}^{mem}B_z I_z^{mem} \\
&\quad + \tilde{\gamma}^{tar}B_z \sum_{samp} I_z^{tar} + S_z A_{zz}^{mem} I_z^{mem} \\
&\quad + S_z \sum_{samp} A_{zz}^{tar} I_z^{tar}.
\end{aligned} \qquad (8)$$

In Eq. (8) the spin operator for the sensor is described by $S_z$ and the memory and target spin operators by $I_z^{mem}$ and $I_z^{tar}$, respectively, where the $z$ axis coincides with the static magnetic field direction and the NV center symmetry axis. Furthermore, the gyromagnetic ratios divided by $2\pi$ are given by the respective $\tilde{\gamma}$ (i.e., $\tilde{\gamma}^{sens} = 28\,GHz\,T^{-1}$, $\tilde{\gamma}^{mem} = 3.08\,MHz\,T^{-1}$, $\tilde{\gamma}^{tar} = 10.7\,MHz\,T^{-1}$) and the crystal-field splitting of the NV sensor-spin triplet ($S = 1$) is denoted by $D = 2.87\,GHz$. Finally, we account for the hyperfine coupling of the sensor to the memory and target spins only via the respective longitudinal coupling constants $A_{zz}$. The latter are $A_{zz}^{mem} = -2.16\,MHz$ for the memory and around $A_{zz}^{tar} \sim 1\,kHz$ for the target spins. The latter follows from the secular approximation and the fact that the symmetry axis of the $^{14}N$ hyperfine tensor is collinear with the $z$ axis and couplings to target spins are expected to be much smaller than their nuclear Zeeman energies.

The NV electron spin has a longitudinal-relaxation time constant of $T_1^{sens} \approx 6$ ms at room temperature, which is in good agreement with literature values[27]. We attribute $T_1^{sens}$ to the time constant of polarization decay in $m_S = 0$ and appearance in $m_S = \pm 1$ after initialization into $m_S = 0$. The difference of both signals is shown in the lower orange line in Fig. 6b. The decay of the NV electron-spin polarization limits the lifetime of coherence of any superposition state of this electron spin. In addition, all other spins in the vicinity of the electron spin, which usually couple like $\propto S_z I_z$, will experience decoherence.

The coherence lifetime of the electron-spin superposition state $|0\rangle + |-1\rangle$ is measured to be $T_2^{sens} = 688\,\mu s$ via a spin echo measurement (see orange line in Fig. 6b). This value determines the limit for coherent phase accumulation in any measurement scenario. It does, however, not limit any storage time of phases. Rather, the coherence lifetime limits the access to strongly coupled nuclear spins (i.e., coupling $> 1/T_2^{sens}$).

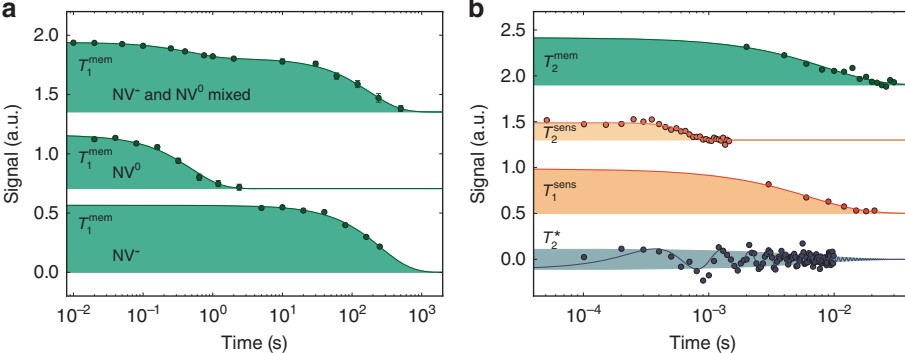

**Fig. 6** Decay constants of sensor, memory, and target. **a** Lifetime of the memory spin. The uppermost graph shows the $T_1^{\text{mem}}$ lifetime at 1.35 T. The data show two different decay constants, 0.49 and 180 s. The fast decay corresponds to those measurement runs, during which the NV center resides in the $NV^0$ charge state, the slow decay corresponds to the $NV^-$ case. By using charge state readout (see ref. [37]) and data post selection, one can isolate pure $NV^0$ and $NV^-$ measurements. The results for 1.5 T are shown in the middle ($NV^0$) and lower ($NV^-$) line. **b** The sensor performance, including the coherence of the probed spins, without any dissipative decoupling. The uppermost line shows the coherence decay of the memory spin, with a decay constant of $T_2^{\text{mem}} = (8.6 \pm 1.3)$ ms. Below, the orange lines show (from top to bottom) the coherence ($T_2^{\text{sens}} = 688 \pm 31$ μs) and longitudinal relaxation ($T_1^{\text{sens}} = 6.4 \pm 0.6$ ms) of the sensor spin. The black line shows the coherence of the target spins (without any decoupling from sensor dissipation), with a decay constant of $T_2^* = 6.78 \pm 1.3$ ms

The $^{14}$N nuclear spin is very isolated in the diamond lattice in a sense that no other nitrogen spin (i.e., $^{14}$N) is available. The $^{14}$N spin is most strongly coupled to the NV electron spin via the hyperfine interaction $H_{\text{hf}} = \mathbf{SAI}$ with tensor $\mathbf{A}$ and spin operator vectors $\mathbf{S} = (S_x, S_y, S_z)$ and $\mathbf{I} = (I_x, I_y, I_z)$. Coupling to other electron or nuclear spins is negligible. In Hamiltonian (8), we have neglected all parts of the hyperfine tensor except for $A_{zz}$.

The memory spin's $T_1^{\text{mem}}$ lifetime is measured for three different settings, namely, for $NV^-$ under dark conditions, for $NV^0$ in the dark and for the case of continuous single-shot readout of the memory spin state[9, 14, 32]. The memory $T_1^{\text{mem}}$ lifetime during single-shot readout scales almost quadratically with increasing magnetic field starting from around 50 mT[32] and reaches a value of around 1 s at 3 T (see fit: $T_1^{\text{mem}}\big|_{\text{SSR}} = 0.155(2) \text{ s}[(B - 50 \text{ mT})\text{T}^{-1}]^{1.68(1)}$ in Fig. 1c). The lifetime during single-shot readout depends on the actual timing of MW, wait and laser pulses during readout (cf. Fig. 1e). Therefore, we also provide the lifetime in terms of laser pulses (300 ns duration each) on the right vertical axis for comparison. The quadratic magnetic field scaling of the $T_1^{\text{mem}}$ lifetime also holds for $NV^0$ in the dark, it reaches around 2 s at 3 T (see fit: $T_1^{\text{mem}}\big|_{\text{NV0}} = 0.239(3) \text{ s T}^{-2} \times B^2$ in Fig. 1c).

The scaling is different for the negative NV center without illumination. The $T_1^{\text{mem}}$ lifetime first increases but then levels off at a value of $260 \pm 20$ s at 1.5 T. We tentatively describe this behavior as resulting from two dissipative rates, a magnetic-field-dependent and a constant one (see fit: $T_1^{\text{mem}}\big|_{\text{NV}-} = \left\{\left[1.7(1.2) \times 10^3 \text{ s T}^{-2}(B - 0.1 \text{ mT})^2\right]^{-1} + [260(40) \text{ s}]^{-1}\right\}$ in Fig. 1c). Nevertheless, 260 and 2 s for $NV^-$ and $NV^0$, respectively, yield quite useful storage times for classical information.

In the following, we describe the measurement of $T_1^{\text{mem}}$ in the dark for the two charge states $NV^-$ and $NV^0$. In a first measurement, we did not discriminate between the charge states. We perform a single $^{14}$N spin-readout step with a duration of a few ms, which yields either low fluorescence for $m_I = +1$ or high fluorescence for $m_I = 0, -1$. We repeat these readout steps until the outcome is "low". In the latter case we switch off the laser illumination via the AOM for a time $\tau$. For $\tau$ larger than 5 s, we additionally flip a beam block into the laser path to avoid laser leakage through the AOM. After time $\tau$, we perform another readout step and record the result. As we did not discriminate the charge states, we do see two decay processes in the upper curve of Fig. 6a.

In a second measurement, we do discriminate between charge states. Therefore, we add a charge state detection sequence before and after step (3). To this end, we send a weak 1 μW orange (594 nm) laser beam for 20 ms and record the fluorescence. In case of $NV^0$, almost only dark counts are detected and in case of $NV^-$ the photon number is considerably higher and thus the charge states can be discriminated[37]. This way we can sort events of $NV^0$ and $NV^-$ lifetime measurements (see the two lower curves in Fig. 6a). In addition, we can check whether the charge state changes during the dark period. We did not see any hint for the latter charge state changes in our measurements on the given timescale.

In the following we consider reasons for the $^{14}$N $T_1^{\text{mem}}$ measurement results. The longitudinal relaxation is not affected by mutual cross-relaxation with other nitrogen spins due to absence of the latter. Minor relaxation rates arise from spin state mixing with the sensor spin via the transverse parts of the hyperfine coupling tensor $\mathbf{A}$ (i.e., $A_\perp \sim 4$ MHz). The mixing $\epsilon$ $\left(|m_S, m_I\rangle = \sqrt{1-\epsilon}|-1, 0\rangle + \sqrt{\epsilon}|0, -1\rangle\right)$ is suppressed by the strong detuning of sensor and memory spin resonances due to the sensor Zeeman energy ($E_{\text{eZ}} \sim 40$ GHz), i.e., $\epsilon \sim 10^{-8} \propto (A_\perp/E_{\text{eZ}})^2$. The mixing in combination with sensor-spin projection event into an $m_S$ eigenstate leads to a flip of the memory spin with probability $\epsilon$.

If the NV center resides in the dark and in its negative charge state, the sensor-spin flips with a rate of ~1 kHz. Therefore, the expected limit on the memory spin $T_1^{\text{mem}}$ lifetime is on the order of $10^5$ s. In the neutral charge state, the electron-spin flip rate is on the order of ~1 MHz and thus we expected memory spin $T_1^{\text{mem}} \sim 100$ s. During single-shot readout of the memory spin, frequent optical excitations lead to stronger mixing because of $A_\perp \approx 40$ MHz in the excited state. In addition, the sensor spin is projected much more often (~1 MHz). Thus the $^{14}$N lifetime during single-shot readout is expected to be ~1 s.

Whereas the memory spin $T_1^{\text{mem}}$ lifetime during single-shot readout is in agreement with our estimates, the lifetimes under dark conditions and in the negative and neutral charge state are much smaller than expected (Fig. 1c), however, the expected ordering of lifetimes for the different conditions is reproduced. One yet unconsidered reason for $^{14}$N $T_1^{\text{mem}}$ relaxation processes might be the quadrupole moment of the nucleus, which couples to electric field gradients.

Because of the particularly strong coupling of the memory to the sensor compared to that of the target spins, the $T_1$ influence of the sensor spin can not be suppressed in our experimental scheme.

Theoretically, the coherence decay constant $T_2^{\text{mem}}$ is expected to be $3/2 \times T_1^{\text{sens}}$ if only affected by the sensor spin $T_1^{\text{sens}}$. Indeed we obtain $T_2^{\text{mem}} = 8.6 \pm 1.4$ ms, which is longer than $T_1^{\text{sens}} \times (T_2^{\text{mem}} = 1.35 T_1^{\text{sens}})$. Similar values have been obtained in ref. [56]. Other influences that might lead to a deviation from the theoretical expectation are laser leakage accompanied by additional sensor-spin decay rates into $m_S = 0$.

Here, the measurement of $T_2^{\text{mem}}$ is performed via a correlation measurement as discussed in ref. [9]. Hence, we have created an equal superposition state on the memory. Then, we have correlated the phase of the memory with the current magnetic field via two sensor CNOT gates separated by $\tau = 1$ μs. The following free evolution time was swept and is displayed as horizontal axis in Fig. 6b. Finally, another pair of sensor CNOT gates separated by $\tau$ is applied, before the phase of the memory is read out (see green line Fig. 6b). The result demonstrates that also coherent metrology information can be stored beyond $T_1^{\text{sens}}$.

**Theoretical derivation of sensor relaxation effects on memory and target spins**. In this section, we are going to derive the effect of sensor relaxation on memory and target spins using a master equation approach.

According to Eq. (8) back-action from the sensor spin on memory and target spins is only mediated via a $A_{zz}S_zI_z$ coupling term. Thus, stochastic flips of the sensor spin (i.e., $T_1^{\text{sens}}$ decay) will lead to decoherence of memory and target spins (i.e., $T_2^*$ and $T_2^{\text{mem}}$ decay) due to an unknown phase accumulation.

Here, we concentrate on the situation of device and target spins in the dark. Relaxation effects can be modeled by investigating a master equation for the combined system of sensor and memory spin with density matrix $\rho = \sum_k \rho_{e,k} \otimes \rho_{n,k}$,

$$\dot{\rho}(t) = -i2\pi[H, \rho(t)] + \sum_j L_j \rho L_j^\dagger - \frac{1}{2}\left(L_j^\dagger L_j \rho + \rho L : j^\dagger L_j\right), \quad (9)$$

with the Lindblad operators $L_j$ describing the stochastic flips of the sensor spin. Given the quasi infinite temperature in the current experiment (i.e., $E/k_B T = 0.014$) the latter decay can be modeled via the depolarizing quantum operation $\mathcal{E}(\rho_e) = \left(1 - \frac{\Delta t}{T_1^{\text{sens}}}\right)\rho_e + \frac{\Delta t}{T_1^{\text{sens}}}\frac{1_e}{3}$ where $\rho_e$ is the sensor-spin density operator. An exemplary

operator sum representation is

$$\mathcal{E}(\rho_e) = \sum_k E_k \rho_e E_k^\dagger$$
$$= \left(1 - \frac{\Delta t}{T_1^{\text{sens}}}\right)\rho_e + \frac{\Delta t}{3T_1^{\text{sens}}} \sum_{n,m=-1}^{1} |n\rangle\langle m|\rho_e|m\rangle\langle n| \quad (10)$$

describing a time-step $\Delta t \ll T_1^{\text{sens}}$ and obeying $\sum_k E_k^\dagger E_k = \mathbf{1}$. Lindblad operators for the depolarizing quantum operation in Eq. (10) are $L_{j=3m+n+5} = \frac{1}{\sqrt{3T_1^{\text{sens}}}}|m\rangle\langle n| \otimes \mathbf{1}_n$ with $m$ and $n$ being the sensor-spin projections $m_S = -1, 0, 1$ and hence $j = 1, .., 9$. This quantum operation reproduces the $T_1^{\text{sens}}$ relaxation of the sensor-spin state polarization. The concomitant sensor-spin decoherence is irrelevant in what follows and therefore does not need to be reproduced. The decoherence is anyway also affected by other sources such as paramagnetic impurities in diamond or the nuclear spin bath.

Next, we solve Eq. (9) numerically with initial state $\rho(0) = |0\rangle\langle 0| \otimes |x\rangle\langle x|$ and observe the remaining coherence via $\text{Tr}\{\text{Tr}_e[\rho(t)]\sigma_x\}$. When varying the coupling term $A_{zz}$, we observe two different regimes (Fig. 4d). For $A_{zz} > 1/T_1^{\text{sens}}$ (e.g., as for the memory qubit) the nuclear spin-coherence time $T_2^* = 3T_1^{\text{sens}}/2$, and for $A_{zz} \ll 1/T_1^{\text{sens}}$ (e.g., as for weakly coupled target spins) $T_2^*$ grows as $A_{zz}^{-2}$.

The first regime (i.e., $A_{zz} > 1/T_1^{\text{sens}}$) can be explained by considering the action of the depolarizing quantum operation for initial state $m_S = 0$. The initial decay out of $m_S = 0$ happens with rate $2/3T_1^{\text{sens}}$, whereas the steady state $\mathbf{1}_e$ is reached with rate $1/T_1^{\text{sens}}$ (Eq. (10)). For couplings $A_{zz} > 1/T_1^{\text{sens}}$ nuclear spin decoherence happens instantly (i.e., in $\Delta t \ll T_1^{\text{sens}}$) upon a transition from $m_S = 0$ into $m_S = \pm 1$. Hence, the amount of nuclear spin-coherence is a measure for the probability that not a single sensor-spin flip has yet occurred, whereas the population of $m_S = 0$ is influenced by rates from and towards $m_S = 0$.

For the second regime (i.e., $A_{zz} T_1^{\text{sens}} \ll 1$), the sensor spin adds tiny additional random phases $\delta\phi \sim A_{zz} T_1^{\text{sens}} m_S \ll 1$ to a nuclear spin superposition state between subsequent sensor-spin flips. The phase uncertainty grows with time $t$ as $\sigma_\phi \propto \delta\phi \sqrt{t/T_1^{\text{sens}}}$. Hence, the dephasing time $T_2^*$ of the nuclear spin scales as $T_2^* \propto T_1^{\text{sens}}/\delta\phi^2 \propto (T_1^{\text{sens}})^{-1} A_{zz}^{-2}$ as expected for the motional averaging regime.

For the NV center in the negative charge state (NV$^-$), we obtain a nuclear spin dephasing time of around 9 ms for couplings down to $A_{zz} \sim 50$ Hz, which corresponds to a sensor–proton-spin distance of ~15 nm (Fig. 4d). For smaller couplings, the nuclear spin $T_2^*$ shows the quadratic increase with inverse coupling constant as derived above.

For the NV center in the neutral charge state (NV$^0$) the setting is different. In its ground state, it features an orbital and an electron-spin doublet[57]. Previous experiments revealed fast decoherence of the $^{14}$N memory spin under these circumstances[36], which can be modeled by an electron spin $S = 1/2$ with a lifetime $T_1^{\text{NV0}} \approx 13$ µs. Given $T_1^{\text{NV0}}$, we can simulate the effect on the $T_2^*$ coherence lifetimes of nuclear spins with varying coupling. The resulting behavior is displayed as the red line in Fig. 4d. For couplings stronger than $\approx 50$ kHz, the nuclear spin lifetime is limited to $\approx 20$ µs and for smaller couplings we see the mentioned quadratic increase. Interestingly, for couplings smaller than ~1.7 kHz nuclear spin $T_2^*$ lifetimes for the NV$^0$ case do overtake the lifetimes for the NV$^-$ case.

Apart from investigating the nuclear spin-coherence for both charge states separately, we are also interested in the behavior during illumination. To model the illumination behavior of the NV$^-$ charge state, we add an additional metastable state to the electronic level system. Illumination and reinitialization of the electron spin is facilitated by spin state-dependent rates into the metastable state, as well as different decay rates back into the different spin states. However, illumination also opens up the door towards ionization of the NV center into the NV$^0$ charge state. As NV$^0$ can have deleterious effects on our classical memory, we want to investigate the case of continuous repolarization without ionization. Ionization events are therefore modeled as instantaneous decoherence on our target spins (Fig. 7). The ionization rates $\gamma_{\text{ion}}$ depend quadratically on the excitation rate $\gamma_{\text{exc}}$ (as it is a two photon process, see ref. [37]).

$$\gamma_{\text{exc}} = P_{\text{Laser}} \times c_{\text{exc}}, \quad (11)$$

$$\gamma_{\text{ion}} = (\gamma_{\text{exc}})^2 \times c_{\text{ion}}. \quad (12)$$

We can numerically model the experimentally obtained $T_2^*$ times for increasing laser power and constant coupling. To this end, we have to adjust parameter such as $c_{\text{exc}}$ and $c_{\text{ion}}$, as well as $p_{\text{branching}}$ the probability of the metastable state to decay into the $m_s = 0$ sublevel, to fit the simulated results to the experimental data (Fig. 3a). We obtain the values $c_{\text{exc}} = 2.5 \times 10^5$ excitations s$^{-1}$ W$^{-1}$, $c_{\text{ion}} = 1/(7 \times 10^8)$ ionizations excitations$^{-2}$ and the branching ratio $p_{\text{branching}} = 0.96$. We observe a tiny initial decrease in $T_2^*$ time, where the increase in optically induced dissipation rate is smaller than the coupling. When increasing the excitation rate beyond the coupling strength, we see an increase due to a sufficiently high and increasing dissipation rate. The $T_2^*$ increase continues towards a global maximum of that model. For higher excitation rates, ionization events destroy access to the memory.

With the optimized model, we can extrapolate the optimum $T_2^*$ time for decreasing target spin-coupling strength. In Fig. 4d, we see the resulting increase of

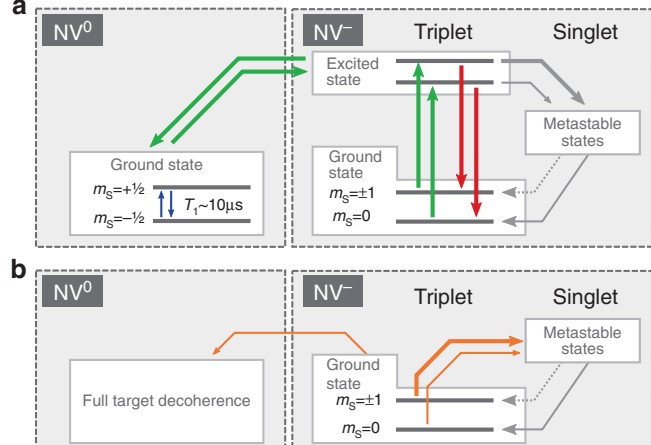

**Fig. 7** NV level schemes for simulation. **a** Level scheme showing the relevant parts of negative and the neutral charge state of the NV center. In NV$^-$, the two spin sub-levels show different fluorescence intensities, as well as a spin polarization by an inter system crossing (ISC) into the metastable singlet state (here drawn as a "black box"). Ionization occurs as an effective two photon process from the NV$^-$ ground state[37], ending up in the uncharged NV$^0$ with a high-spin dissipation rate. Recombination also occurs as a two photon process. For the simulation of the pure NV$^-$ and NV$^0$ case, only the respective ground states are modeled. **b** The reduced level scheme for the simulation of the continuous excitation case. The NV$^-$ excited state is omitted due to its short lifetime. The spin-dependent ISC is realized by spin-dependent excitation rates. Ionization to the NV$^0$ charge state is implemented by immediate target spin decoherence

$T_2^*$ with a slightly smaller slope as for the NV$^0$ and NV$^-$ case in the dark. Of course, the model fits to our measurement results in the kHz coupling range. Next, we check for consistency of the further increase.

As in the dark cases, the coherence time scales as $T_2^* \propto \Gamma/A_{zz}^2$ (Eq. (2)), where $\Gamma \propto \gamma_{\text{exc}}$ (Eq. (11)). However, at the optimum, $T_2^*$ is equally limited by the ionization rate as $T_2^* \propto 1/\gamma_{\text{ion}} \propto 1/\Gamma^2$ (Eq. (12)), and therefore scales quadratically with the inverse optical excitation rate, as with the dissipation rate. Hence, we require $\Gamma/A_{zz}^2 \propto 1/\Gamma^2$ (i.e., $A_{zz}^2 \propto \Gamma^3$) leading to a coherence time scaling of $T_2^* \propto A_{zz}^{-4/3}$. This result perfectly agrees with the numerically simulated scaling in Fig. 4d.

**Data availability.** Data supporting the findings of this study are available within the article and its Methods section, and from the corresponding authors upon reasonable request.

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

## Acknowledgements

We thank Sebastian Zaiser, Ville Bergholm, Nikolas Abt, Ingmar Jakobi, Johannes Greiner, Florian Dolde, Marcus Doherty, and Fedor Jelezko for fruitful discussions and technical advice. We acknowledge financial support by the German Science Foundation (SFB-TR 21, SFB 716, and SPP1601), the EU (DIADEMS), the Volkswagen Stiftung, the JST, and JSPS KAKENHI (Nos. 26246001 and 26220903), and the National Science Foundation (NSF-1401632), and Research Corporation.

## Author contributions

M.P., P.N., C.A.M., P.N., and J.W. conceived the robust memory-assisted correlation spectroscopy technique and the applied dissipative-decoupling methods. M.P. and N.A. performed all single NV measurements. M.P., N.A., and P.N. analyzed and discussed the measurement results. H.S., S.O., and J.I. conceived and conducted the synthesis and fabrication of suitable NV diamond substrate. All authors participated in manuscript preparation. P.N. and J.W. supervised the project.

## Additional information

**Competing interests:** The authors declare no competing financial interests.

