## [Peer Review File · Nature Communications]

Reviewers' Comments:

Reviewer #1:

Remarks to the Author:

%Nonvolatile quantum memory enables sensor unlimited nanoscale spectroscopy of finite nite quantum systems

In this paper the authors describe and experimentally demonstrate a protocol for achieving a tenfold improvement in the nuclear spin sensing sensitivity using single NV center in diamond. This is achieved by using an ancillary nitrogen nuclear spin as a memory to partially store phase information during two separate interrogation steps. Further, the authors show that this memory is robust enough such that the electronic sensor spin and the target spin can be decoupled during the waiting time by optical pumping or even ionization techniques. This suppresses the sensor induced dephasing of the target spin. Both techniques combined allow very narrow linewidths, which are no longer limited by the inverse of the T1 time of the electron spins.

The experimental demonstration of a tenfold improvement of the resolution of NV-based nanoscale NMR spectroscopy is certainly a nontrivial achievement, given the already quite advanced stage of this field, and also an important step towards real chemistry applications. The authors combine a several experimental techniques and identify an optimal combination to achieved a real spectroscopic benefit and I think that many of the insights presented here will be very important for a further advancement of the field. The results and conclusions of this work are well supported by the presented data and the extensive analysis given in the supplementary material. Overall this is a very professional paper and I recommend it for publication in Nature Communications.

My only minor complaint about this work is that it is very technical and hard to follow for readers who are not immediately working at the forefront of the field. In particular, already in the introduction one gets easily lost with all the definition of T₁ and T₂ times for three different types of spins (sensor, target, memory), which are not properly introduced at this stage. Also the concepts of a classical memory or that of a "in-situ correlation spectroscopy measurement" are not commonly known and maybe these concepts could be further illustrated by explicit formulas or illustrative graphs. However, I leave this up to the authors.

Reviewer #2:

Remarks to the Author:

The manuscript by Matthias Pfender et al. investigates the NV electron spin - host nuclear spin system for high spectral resolution sensing and demonstrates techniques to decouple the sensor spin from both the host nuclear spin and the target spin. Using these techniques they achieve spectral linewidths of 13 Hz, which exceed previous reports by an order of magnitude, bringing them closer to being able to achieve linewidths of a few Hz required for NV-based NMR.

The dissipative decoupling techniques shown in this paper are novel in this particular realisation and I believe the results to be of interest to researchers in the community and in the wider field. Their exploration and analysis of different decoupling regimes sheds light on the mechanisms at play and indicates which technique is the most beneficial for a particular target spin. However, I think that the title and the main text don't reflect their achievement, but instead place too much emphasis on the N14 as a quantum memory, which is not a novel idea and has already been demonstrated multiple times.

In a reviewed manuscript I would like to see the following criticisms addressed:

1) The title doesn't reflect what is new it this work. Especially the word 'nonvolatile' and 'sensor

unlimited' do not describe the achievements of this paper.

- Nonvolatile: The N14 spin is still very much a volatile memory, as demonstrated by its finite T1. The word 'nonvolatile' suggests a new kind of memory, but the 14N is already being used as a memory in multiple groups.

- Sensor unlimited: the linewidth is still very much limited by sensor T1 as can be seen in Fig 6 b (T1 sens limits T2* target), 'sensor unlimited' is therefore misleading.

2) In the abstract the authors claim to use the N14 spin as a quantum and classical storage, but no data is shown for quantum storage during the 'storage and target manipulation' part. If the authors mean the encoding part of the scheme to involve quantum storage, this should be specified.

3) The focus on using N14 in the first part of the main text is misleading as it isn't a new technique. It distracts from the novelty of the the 3-part sensing scheme and the decoupling using the NV0 state or using continuous optical yellow illumination.

4) As the authors focus on molecular NMR in the abstract and introduction, I would like to see a discussion of more 'real life' measurements, not detecting spins internal to the diamond lattice but trying to resolve chemical shifts within a molecule. What limits T2 of target spins in that case?

Minor comments:

a) No mention of what open and filled circles mean in Fig. 1d) (in the main text)

b) Check use of 'further' and 'furthermore' in lines 32 and 50, 'deliberately' in line 70.

c) Last paragraph of Introduction presents the concept of using a nuclear spin as a memory for quantum and classical information as the novel result in this work. This has been reported several times already (including by the same group in Nat. Commun 7:12279 (2016)). This memory hasn't changed and only seems to have gained the new name of 'non-volatile memory'. Emphasis in the last paragraph of the introduction should be on the new dissipative decoupling techniques.

d) Figure 1 (d) caption: define 'proper phases'

e) Line 88: 'As a main point of this work we utilise the expectation value of the N nuclear spin...' This is not the main novelty of the work.

f) Line 131: 'less volatile' T1 storage, becomes non-volatile? Not a new idea.

g) Suggestion: Give names to the two 13C atoms, e.g. target A, target B, to avoid referring to spins as in line 171 'previous one' or having to repeat '-1.8 kHz coupled spin' and '2.8 kHz coupled spin' multiple times.

h) Caption Fig. 3: '...', using only the conversion from applied laser power...' Please specify what these parameters are used as, presumably fitting parameters.

i) Change Fig. 4 d) Axis label to 'Target spin T2' from 'Nuclear spin T2' for consistency.

j) Line 198: '... capable of quantum and classical metrology data storage'. True but misleading as authors don't perform quantum storage.

k) Line 199: 'We have developed...' again, misleading. Focus on new aspects of correlation spectroscopy.

Supplementary:

l) B1: How is the memory spin initialised?

m) The authors speak about their three storage steps in the supplementary. It would benefit the clarity of the main text to adopt this description in the main text as well.

n) C1: In line 368 'green' should be 'orange'

o) Figure 6 could benefit from labels for each measurement (such as in Fig. 1b)

Reviewer #3:

Remarks to the Author:

This is an excellent manuscript. I recommend it for publication in Nature Communications. The presented work is very interesting both for the quantum sensing and solid-state quantum

information communities and the manuscript is well written. However, before a decision can be made on publication, the authors should address the following:

1) The manuscript does a good job of making clear what problems are actually being solved/investigated: 1) protecting target spin from sensor spin relaxation and 2) precise spectroscopy by in-situ storage of correlations in the memory spin. However, these topics are presented as very general challenges for sensing/metrology (see e.g. the abstract and the introduction). The actual situation appears to be more subtle in that these challenges appear only under certain conditions. In particular, NV sensor relaxation is only an issue at room temperature and correlations can alternatively be stored classically by reading out the system. In those cases similar or better spectroscopy results than reported here have already been reported. It is clear that the situation the authors consider is intrinsically interesting, but many of the statements made are too general and need to be refined.

First, regarding sensor relaxation, this is not an issue at cryogenic temperatures where T1 times are minutes or hours. Therefore many of the statements throughout the manuscript are only true with the addition of "at room temperature". The authors should add a sentence to the introduction that makes clear that they are limiting the discussion to room temperature (and therefore sensor $T_1 = \sim 6$ ms).

Second, regarding storing correlations, it does not become entirely clear if there is a fundamental advantage of storing the classical correlation in an in-situ memory over reading out the result and storing it on a computer. Given that there is a (single-shot) readout mechanism of the sensor state, high-resolution spectroscopy appears to be available directly by simply sensing + readout, waiting or manipulating the target, then again sensing + readout.

Examples where equal or better spectroscopic resolution was already demonstrated are reference 27 (Figure 3C, spectroscopy of a C13 spin at room temperature, decoupling from sensor relaxation by laser illumination, correlations between consecutive repeated readouts) and Nature Communications 7:11526, 2016 (Figure 1C, spectroscopy of C13 spins, no sensor relaxation due to cryogenic temperature, direct correlations between single-shot readouts of sensor spin). The authors should revise the manuscript to reflect that equal or better spectroscopic resolution has already been performed in the mentioned papers, but using alternative methods under different conditions.

Again, these comments do not compromise the presented results in any way. But they do seem to challenge the validity of some of the very broad general statements made throughout the manuscript. Here are some examples (from the abstract and beginning of the manuscript):

"The dissipation of nitrogen-vacancy centers in diamond is sufficiently small for detection of individual or small ensembles of nuclear spins, yet limiting spectral resolution of nuclear magnetic resonance (NMR) spectroscopy to a few hundred Hz" → only true at room temperature and if no decoupling of sensor (for example, this statement disregards reference 27).

"NV centers enable target spin identification with spectral linewidths of several hundred Hz" – much better linewidths were already demonstrated in the above-mentioned works.

"In previous NV-based NMR experiments the achievable linewidth was limited by the finite relaxation time of the sensor..." – not true for experiments at cryogenic temperatures or previous experiments with decoupling of sensor/target (for example this statement disregards reference 27).

"The longitudinal relaxation time of the NV center sensor spin's expectation value ... of ... 6ms is usually one to three orders of magnitude longer than its transverse relaxation time (see Fig. 1b)." – this value is only accurate at room temperature.

2) I am a little confused about the title, which seems inaccurate.

- "Nonvolatile quantum memory" the main idea in this paper is to use the Nitrogen nuclear spin as a classical memory, not as a quantum memory. The word quantum should be removed here.

- "of finite quantum systems". It is not clear to me what is meant with finite quantum systems, and this does not become clear from the text. Please explain.

General remarks

We thank the referees for carefully reviewing our manuscript.

First we will give a few general comments before we give point by point answers to the raised issues and questions. Typically, our answers are shown in blue, whereas the reviewer comments are displayed in black. Finally, we will attach a highlighted version of the manuscript showing in red all changes to the previous version. In this document you will find a separate reference list too follow our arguments more easily.

Although our work was appreciated, the reviewers had three main concerns.

First, our manuscript was lacking a general motivation for applying correlation spectroscopy with NV centers at room temperature. We have added a required description.

Second, there were doubts about the novelty of using the ^{14}N nuclear spin as classical memory. We have now emphasized that this is indeed the first work, where the robustness of the ^{14}N spin under free evolution, during readout, during sensor reinitialization and during decoupling was demonstrated. We have analyzed the scaling of this robustness for a huge magnetic field range. Consequently, it is also the first work, where classical information was stored beyond the T_1^{sens} -limit of the sensor. Previous work used volatile ^{14}N quantum memories [Zaiser et al., 2016], used classical ^{14}N memory but not in “nonvolatile mode” as it was derived in this work [Laraoui et al., 2013b] or simultaneously presented similar results on the arXiv [Roskopf et al., 2016] compared to ours [Pfender et al., 2016]. We have emphasized in the revised manuscript, what our achievements towards a robust classical memory are.

Third, we have presented our work in a too general manner while not emphasizing other relevant work. In particular we have not explicitly made clear that many given facts are valid only for NV centers at room temperature. We emphasize our room temperature operation mode, now. In addition, we present results from cryogenic experiments and show its distinction. Furthermore, we did not highlight the paper by Maurer et al. [2012] enough, where to date the smallest linewidth of a single ^{13}C spin was demonstrated, again employing decoupling as we did. Now, we discuss this paper in more detail and show what are the benefits using our novel method.

Reviewers' comments:

Reviewer #1 (Remarks to the Author):

In this paper the authors describe and experimentally demonstrate a protocol for achieving a tenfold improvement in the nuclear spin sensing sensitivity using single NV center in diamond. This is achieved by using an ancillary nitrogen nuclear spin as a memory to partially store phase information during two separate interrogation steps. Further, the authors show that this memory is robust enough such that the electronic sensor spin and the target spin can be decoupled during the waiting time by optical pumping or even ionization techniques. This suppresses the sensor induced dephasing of the target spin. Both techniques combined allow very narrow linewidths, which are no longer limited by the inverse of the T_1 time of the electron spins.

The experimental demonstration of a tenfold improvement of the resolution of NV-based nanoscale NMR spectroscopy is certainly a nontrivial achievement, given the already quite advanced stage of this field, and also an important step towards real chemistry applications. The authors combine a several experimental techniques and identify an optimal combination to achieved a real spectroscopic benefit and I think that many of the insights presented here will be very important for a further advancement of the field. The results and conclusions of this work are well supported by the presented data and the extensive analysis given in the supplementary material. Overall this is a very professional paper and I recommend it for publication in Nature Communications.

My only minor complaint about this work is that it is very technical and hard to follow for readers who are not immediately working at the forefront of the field. In particular, already in the introduction one gets easily lost with all the definition of T_1 and T_2 times for three different types of spins (sensor, target, memory), which are not properly introduced at this stage. Also the concepts of a classical memory or that of a "*in-situ correlation spectroscopy measurement*" are not commonly known and maybe these concepts could be further illustrated by explicit formulas or illustrative graphs. However, I leave this up to the authors.

We have tried to improve the introduction regarding general accessibility, furthermore we have added a paragraph motivating *in-situ correlation spectroscopy*. In the remaining text we tried to find a balance between technical relevant details and clarity.

For proper definition and overview of timescales we have added a table.

Reviewer #2 (Remarks to the Author):

The manuscript by Matthias Pfender et al. investigates the NV electron spin - host nuclear spin system for high spectral resolution sensing and demonstrates techniques to decouple the sensor spin from both the host nuclear spin and the target spin. Using these techniques they achieve spectral linewidths of 13 Hz, which exceed previous reports by an order of magnitude, bringing them closer to being able to achieve linewidths of a few Hz required for NV-based NMR.

The dissipative decoupling techniques shown in this paper are novel in this particular realisation and I believe the results to be of interest to researchers in the community and in the wider field. Their exploration and analysis of different decoupling regimes sheds light on the mechanisms at play and indicates which technique is the most beneficial for a particular target spin. However, I think that the title and the main text don't reflect their achievement, but instead place too much emphasis on the ^{14}N as a quantum memory, which is not a novel idea and has already been demonstrated multiple times.

In a reviewed manuscript I would like to see the following criticisms addressed:

- 1) The title doesn't reflect what is new it this work. Especially the word '*nonvolatile*' and '*sensor unlimited*' do not describe the achievements of this paper.
 - Nonvolatile: The ^{14}N spin is still very much a volatile memory, as demonstrated by its finite T_1 . The word '*nonvolatile*' suggests a new kind of memory, but the ^{14}N is already being used as a memory in multiple groups.

Below we answer the referee’s criticism of using the word *nonvolatile*.

First, we motivate the use of the word *nonvolatile* by the fact that it survives much longer than any time scale relevant in our measurements. Furthermore, nonvolatile also includes a certain robustness to various external influences. As pointed out in Figure 1c and partially anticipated from previous work (e.g. [Neumann et al., 2010]) we make the memory robust by going to higher magnetic fields, here > 1 T. Nonvolatile refers to several aspects:

- (i) Classical information is stored well beyond all other relevant time scales in the experiment, first demonstrated here.
- (ii) Stored information survives reinitialization of the sensor [Neumann et al., 2010].
- (iii) Stored information survives readout of the memory [Neumann et al., 2010].
- (iv) Stored information survives dissipative decoupling of target from the sensor, first demonstrated here.

Second, regarding novelty, we characterize and use the ^{14}N spin as robust/nonvolatile memory for the first time. For comparison, in previous experiments, it has been used as quantum and therefore explicitly volatile memory [Zaiser et al., 2016]. The ^{14}N spin was also previously used as classical memory [Laraoui et al., 2013a] but then it has neither been proven to be robust nor operated in robust mode, as we know today. In fact, the latter paper does speculate about their achievable spectral resolution of 200 Hz, which has been achieved recently even with a volatile quantum (not classical) memory in [Zaiser et al., 2016]. A similar work, also using the ^{14}N spin as classical memory, was presented on the arXiv one week before ours (compare [Pfender et al., 2016, Roskopf et al., 2016]). Therefore, we still think that investigating and demonstrating the NV intrinsic nitrogen spin as robust classical memory is indeed an appropriate achievement of our work, which deserves to be highlighted.

- Sensor unlimited: the linewidth is still very much limited by sensor T_1 as can be seen in Fig 6 b (T_1^{sens} limits $T_2^{*,\text{target}}$), ‘*sensor unlimited*’ is therefore misleading.

We see that figure 6b and the accompanying caption are misleading. Although, there is the comment that the presented decaying oscillation is for the non-decoupling case, in the end we give the hint that T_2^* is limited by T_1^{sens} , omitting the necessary statement that this is only true without decoupling. We have improved the caption.

Nevertheless, figure 4d illustrates the limit of our decoupling techniques. Indeed, we are not completely sensor-unlimited, however, much less limited. Our experiments yield an increase of the sensor limit by a factor of four for coupling strengths of about 2kHz, at a distance of up to 3nm. When we extrapolate our technique to single nuclear spins at the coherent interaction limit the improvement is already 20-fold. Current, nanoscale NMR experiments with target spin distances of about 10 nm would benefit from two orders of magnitude improvement. The latter limit is then beyond one second, i.e. beyond the free evolution times in conventional NMR experiments. Therefore, we think it is fair to say that our technique “*enables sensor-unlimited spectroscopy*”.

We now make clear in the text what we mean by the statement “*sensor-unlimited*”.

- 2) In the abstract the authors claim to use the ^{14}N spin as a quantum and classical storage, but no data is shown for quantum storage during the ‘*storage and target manipulation*’ part. If the authors mean the encoding part of the scheme to involve quantum storage, this should be specified.

The current measurement scheme exploits both, the quantum and the classical memory aspect of the ^{14}N nuclear spin. Indeed, the referee is right that the gain in spectral resolution is mainly due to the classical storage aspect during the “*storage and target manipulation part*”. We have added a more detailed explanation of the measurement sequence, saying that quantum storage only occurs during short intervals during encoding and decoding and that during storage and manipulation we use the classical memory aspect of our nitrogen spin. Note that quantum storage during the encoding and decoding parts is indeed a benefit

because (i) it increases sensitivity [Zaiser et al., 2016] and (ii) it is faster due to lack of additional RF pulses for quantum \leftrightarrow classical memory switching. To avoid confusion we have also changed the title.

- 3) The focus on using ^{14}N in the first part of the main text is misleading as it isn't a new technique. It distracts from the novelty of the the 3-part sensing scheme and the decoupling using the NV^0 state or using continuous optical yellow illumination.

As pointed out above (points about robustness), we think that our demonstrated utilization of the ^{14}N nuclear spin is indeed novel. Actually, the presented robustness enables the 3-part sensing scheme including decoupling in the first place. Therefore, we think it deserves to be mentioned. Furthermore, we have added more details about the decoupling techniques to the introduction.

However, we agree that more distinction to previous experiments is necessary. We have improved the text accordingly.

- 4) As the authors focus on molecular NMR in the abstract and introduction, I would like to see a discussion of more 'real life' measurements, not detecting spins internal to the diamond lattice but trying to resolve chemical shifts within a molecule. What limits T_2 of target spins in that case?

Indeed, the current measurements are very much different from NMR on external nuclear spins. The intrinsic ^{13}C spins of the diamond lattice are pretty isolated stationary spins. There are existing proposals to use them as a resource for qubits or as a quantum simulation testbed [Cai et al., 2013]. The latter is another application of the demonstrated method. Hence, we do not exclusively want to concentrate on NMR of external spins. Typically, in previous NV-based NMR experiments with external spins the limiting factor for NMR linewidth has been for example diffusion of target spins or dipolar broadening but also insufficient sensor coherence time [Staudacher et al., 2013, Mamin et al., 2013, DeVience et al., 2015, Staudacher et al., 2015, Häberle et al., 2015, Kong et al., 2015, Lovchinsky et al., 2016, 2017]. However, it is very likely that all mentioned limitations can be overcome and high-resolution NMR on external nuclear spins can be demonstrated. That's where our novel technique is going to be applied.

We have added a more thorough discussion about potential applications of our method and additional current limits to nano-scale NMR spectroscopy on 'real-life' samples.

Minor comments:

- a) No mention of what open and filled circles mean in Fig. 1d) (in the main text)
This is now explained in the caption.
- b) Check use of 'further' and 'furthermore' in lines 32 and 50, 'deliberately' in line 70.
Done.
- c) Last paragraph of Introduction presents the concept of using a nuclear spin as a memory for quantum and classical information as the novel result in this work. This has been reported several times already (including by the same group in Nat. Commun 7:12279 (2016) [here: [Zaiser et al., 2016]]). This memory hasn't changed and only seems to have gained the new name of '*non-volatile memory*'. Emphasis in the last paragraph of the introduction should be on the new dissipative decoupling techniques.
We have made clear that no demonstration of nitrogen as robust classical memory was presented before and we have added more details about our dissipative decoupling techniques.
- d) Figure 1 (d) caption: define '*proper phases*'
We have added a comment that the phases are given as subscripts x and y in the wire diagram.
- e) Line 88: '*As a main point of this work we utilise the expectation value of the N nuclear spin ...*' This is not the main novelty of the work.
We have altered the corresponding paragraph.

- f) Line 131: '*less volatile*' T_1 storage, becomes non-volatile? Not a new idea.
We have changed the sentence.
- g) Suggestion: Give names to the two ^{13}C atoms, e.g. target A, target B, to avoid referring to spins as in line 171 '*previous one*' or having to repeat '*-1.8 kHz coupled spin*' and '*2.8 kHz coupled spin*' multiple times.
We followed the suggestion and gave names to the target spins, A_1 and A_2 for target spins around NV A and B_1 for the observable target spin around NV B.
- h) Caption Fig. 3: '*... , using only the conversion from applied laser power ...*' Please specify what these parameters are used as, presumably fitting parameters.
We have improved caption 3.
- i) Change Fig. 4 d) Axis label to '*Target spin T_2* ' from '*Nuclear spin T_2* ' for consistency.
We have changed the axis label.
- j) Line 198: '*... capable of quantum and classical metrology data storage*'. True but misleading as authors don't perform quantum storage.
We have changed the summary accordingly.
- k) Line 199: '*We have developed ...*' again, misleading. Focus on new aspects of correlation spectroscopy.
We make clear that we have adapted correlation spectroscopy to support quantum and classical storage on a separate memory spin along with suitable decoupling of target and memory spins from sensor dissipation.

Supplementary:

- l) B1: How is the memory spin initialised?
We have added a comment to the main text and the appendix saying that a readout result reports the initial memory state for the next run.
- m) The authors speak about their three storage steps in the supplementary. It would benefit the clarity of the main text to adopt this description in the main text as well.
We have now mentioned all three storage steps in the main text.
- n) C1: In line 368 '*green*' should be '*orange*'
We have corrected the color reference.
- o) Figure 6 could benefit from labels for each measurement (such as in Fig. 1b)
We have added the same labels to figure 6.

Reviewer #3 (Remarks to the Author):

This is an excellent manuscript. I recommend it for publication in Nature Communications. The presented work is very interesting both for the quantum sensing and solid-state quantum information communities and the manuscript is well written. However, before a decision can be made on publication, the authors should address the following:

- 1) The manuscript does a good job of making clear what problems are actually being solved/ investigated:
 - 1) protecting target spin from sensor spin relaxation and
 - 2) precise spectroscopy by in-situ storage of correlations in the memory spin.

However, these topics are presented as very general challenges for sensing/metrology (see e.g. the abstract and the introduction). The actual situation appears to be more subtle in that these challenges appear only under certain conditions. In particular, NV sensor relaxation is only an issue at room temperature and correlations can alternatively be stored classically by reading out the system. In those cases similar or better spectroscopy results than reported here have already been reported. It is clear that the situation the authors consider is intrinsically interesting, but many of the statements made are too general and need to be refined.

First, regarding sensor relaxation, this is not an issue at cryogenic temperatures where T_1 times are minutes or hours. Therefore many of the statements throughout the manuscript are only true with the addition of “at room temperature”. The authors should add a sentence to the introduction that makes clear that they are limiting the discussion to room temperature (and therefore sensor $T_1 \approx 6$ ms).

We have now clarified the room temperature aspect of our work.

Second, regarding storing correlations, it does not become entirely clear if there is a fundamental advantage of storing the classical correlation in an in-situ memory over reading out the result and storing it on a computer. Given that there is a (single-shot) readout mechanism of the sensor state, high-resolution spectroscopy appears to be available directly by simply sensing + readout, waiting or manipulating the target, then again sensing + readout.

We have added a discussion/comparison about non-in-situ or conventional correlation spectroscopy as described by the reviewer. In particular we have discussed why this is feasible for NV centers at cryogenic temperatures and not at room temperature. We have also mentioned that single shot readout of the sensor (e.g. via the memory, without storage) has been demonstrated to affect target spins. In addition, single shot readout of the NV center is likely to ionize the NV leading to reduced correlation between subsequent measurement results [Waldherr et al., 2011, Aslam et al., 2013]. In general, a non-ideal readout visibility V further reduces the visibility of the conventional correlation signal as V^2 , whereas insitu correlation spectroscopy exhibits visibility scaling as V .

Examples where equal or better spectroscopic resolution was already demonstrated are reference 27 [here: [Maurer et al., 2012]] (Figure 3C, spectroscopy of a ^{13}C spin at room temperature, decoupling from sensor relaxation by laser illumination, correlations between consecutive repeated readouts) and Nature Communications 7:11526, 2016 [here: [Cramer et al., 2016]] (Figure 1C, spectroscopy of ^{13}C spins, no sensor relaxation due to cryogenic temperature, direct correlations between single-shot readouts of sensor spin). The authors should revise the manuscript to reflect that equal or better spectroscopic resolution has already been performed in the mentioned papers, but using alternative methods under different conditions.

Now, we have clearly emphasized the results of [Maurer et al., 2012] in the text. However, we have also mentioned, that this demonstration is not universal, as it requires a particular ^{13}C nuclear spin. Furthermore, it inhibits storage on the nitrogen spin and readout of the demonstrated memory takes about 100 times longer than our readout. We have estimated the time of a basic single shot readout step of $\approx 170 \mu\text{s}$ vs. $\approx 1.5 \mu\text{s}$ in the nitrogen case. A complete readout takes multiples of such a basic step.

We have now added the given example of a cryogenic experiment, where the demonstrated spectral resolution is $\approx (\pi \cdot 18.2 \text{ms})^{-1} = 17.5 \text{Hz}$ (Figure 1C, qubit 3). That is slightly broader than the best value we have presented, however, does not seem to be limited by the applied technique but by the coherence time of the target spin itself.

Again, these comments do not compromise the presented results in any way. But they do seem to challenge the validity of some of the very broad general statements made throughout the manuscript. Here are some examples (from the abstract and beginning of the manuscript):

- “*The dissipation of nitrogen-vacancy centers in diamond is sufficiently small for detection of individual or small ensembles of nuclear spins, yet limiting spectral resolution of nuclear magnetic resonance (NMR) spectroscopy to a few hundred Hz*” — only true at room temperature and if no decoupling of sensor (for example, this statement disregards reference 27 [here: [Maurer et al., 2012]]).
Now, we have restricted our claim to room temperature operation of NV sensors. Apart from that, dissipation of the NV electron spin at room temperature indeed limits resolution in general (i.e. if no decoupling is applied). So we keep this information. As before, however, we explicitly say, just a few sentences later, that we need to employ decoupling. As the abstract is not referenced in *nature communications* we have not given any reference here. Now, however, we have more clearly mentioned the achievements of [Maurer et al., 2012].
- “*NV centers enable target spin identification with spectral linewidths of several hundred Hz*” — much better linewidths were already demonstrated in the above-mentioned works.
We have restricted this statement to room temperature applications. Furthermore, We have considerably revised this part of the manuscript. However, keep in mind that this statement in general is correct for room temperature. Only if you apply a suitable decoupling method you break the sensor limitation, which is what we clearly say.
- “*In previous NV-based NMR experiments the achievable linewidth was limited by the finite relaxation time of the sensor . . .*” — not true for experiments at cryogenic temperatures or previous experiments with decoupling of sensor/target (for example this statement disregards reference 27 [here: [Maurer et al., 2012]]).
We have now incorporated also cryogenic measurement results and we have discussed the achievements of [Maurer et al., 2012]
- “*The longitudinal relaxation time of the NV center sensor spin’s expectation value . . . of . . . 6ms is usually one to three orders of magnitude longer than its transverse relaxation time (see Fig. 1b).*” — this value is only accurate at room temperature.
We have now restricted the insitu correlations spectroscopy discussion to room temperature.

2) I am a little confused about the title, which seems inaccurate.

- “*Nonvolatile quantum memory*” the main idea in this paper is to use the Nitrogen nuclear spin as a classical memory, not as a quantum memory. The word quantum should be removed here.
We have removed the word quantum from the title.
- “*of finite quantum systems*”. It is not clear to me what is meant with finite quantum systems, and this does not become clear from the text. Please explain.
We have changed the title, which now refers to small spin clusters.

References

- N. Aslam, G. Waldherr, P. Neumann, F. Jelezko, and J. Wrachtrup. Photo-induced ionization dynamics of the nitrogen vacancy defect in diamond investigated by single-shot charge state detection. *New J. Phys.*, 15(1):013064, Jan. 2013. ISSN 1367-2630. doi: 10.1088/1367-2630/15/1/013064. URL <http://iopscience.iop.org/1367-2630/15/1/013064>.
- J. Cai, A. Retzker, F. Jelezko, and M. B. Plenio. A large-scale quantum simulator on a diamond surface at room temperature. *Nat. Phys.*, 9(3):168–173, 2013. ISSN 1745-2473. doi: 10.1038/nphys2519. URL <http://www.nature.com/nphys/journal/v9/n3/full/nphys2519.html>.

- J. Cramer, N. Kalb, M. A. Rol, B. Hensen, M. S. Blok, M. Markham, D. J. Twitchen, R. Hanson, and T. H. Taminiau. Repeated quantum error correction on a continuously encoded qubit by real-time feedback. *Nat. Commun.*, 7:11526, 2016. doi: 10.1038/ncomms11526. URL <http://www.nature.com/ncomms/2016/160505/ncomms11526/full/ncomms11526.html>.
- S. J. DeVience, L. M. Pham, I. Lovchinsky, A. O. Sushkov, N. Bar-Gill, C. Belthangady, F. Casola, M. Corbett, H. Zhang, M. Lukin, H. Park, A. Yacoby, and R. L. Walsworth. Nanoscale NMR spectroscopy and imaging of multiple nuclear species. *Nat. Nanotechnol.*, 10(2):129–134, Feb. 2015. ISSN 1748-3387. doi: 10.1038/nnano.2014.313. URL <http://www.nature.com/nnano/journal/v10/n2/full/nnano.2014.313.html>.
- T. Häberle, D. Schmid-Lorch, F. Reinhard, and J. Wrachtrup. Nanoscale nuclear magnetic imaging with chemical contrast. *Nat. Nanotechnol.*, 10(2):125–128, Feb. 2015. ISSN 1748-3387. doi: 10.1038/nnano.2014.299. URL <http://www.nature.com/nnano/journal/v10/n2/full/nnano.2014.299.html>.
- X. Kong, A. Stark, J. Du, L. P. McGuinness, and F. Jelezko. Towards Chemical Structure Resolution with Nanoscale Nuclear Magnetic Resonance Spectroscopy. *Phys. Rev. Appl.*, 4(2):024004, Aug. 2015. doi: 10.1103/PhysRevApplied.4.024004. URL <http://link.aps.org/doi/10.1103/PhysRevApplied.4.024004>.
- A. Laraoui, F. Dolde, C. Burk, F. Reinhard, J. Wrachtrup, and C. A. Meriles. High-resolution correlation spectroscopy of ^{13}C spins near a nitrogen-vacancy centre in diamond. *Nat. Commun.*, 4:1651, Apr. 2013a. doi: 10.1038/ncomms2685. URL <http://www.nature.com/ncomms/journal/v4/n4/full/ncomms2685.html>.
- A. Laraoui, F. Dolde, C. Burk, F. Reinhard, J. Wrachtrup, and C. A. Meriles. High-Resolution Correlation Spectroscopy of ^{13}C Spins Near a Nitrogen-Vacancy Center in Diamond. *Nature Communications*, 4:1651, Apr. 2013b. ISSN 2041-1723. doi: 10.1038/ncomms2685. URL <http://arxiv.org/abs/1305.1536>. arXiv: 1305.1536.
- I. Lovchinsky, A. O. Sushkov, E. Urbach, N. P. d. Leon, S. Choi, K. D. Greve, R. Evans, R. Gertner, E. Bersin, C. Müller, L. McGuinness, F. Jelezko, R. L. Walsworth, H. Park, and M. D. Lukin. Nuclear magnetic resonance detection and spectroscopy of single proteins using quantum logic. *Science*, 351(6275):836–841, Feb. 2016. ISSN 0036-8075, 1095-9203. doi: 10.1126/science.aad8022. URL <http://science.sciencemag.org/content/351/6275/836>.
- I. Lovchinsky, J. D. Sanchez-Yamagishi, E. K. Urbach, S. Choi, S. Fang, T. I. Andersen, K. Watanabe, T. Taniguchi, A. Bylinskii, E. Kaxiras, P. Kim, H. Park, and M. D. Lukin. Magnetic resonance spectroscopy of an atomically thin material using a single-spin qubit. *Science*, 355(6324):503–507, Feb. 2017. ISSN 0036-8075, 1095-9203. doi: 10.1126/science.aal2538. URL <http://science.sciencemag.org/content/355/6324/503>.
- H. J. Mamin, M. Kim, M. H. Sherwood, C. T. Rettner, K. Ohno, D. D. Awschalom, and D. Rugar. Nanoscale Nuclear Magnetic Resonance with a Nitrogen-Vacancy Spin Sensor. *Science*, 339(6119):557–560, Feb. 2013. ISSN 0036-8075, 1095-9203. doi: 10.1126/science.1231540. URL <http://www.sciencemag.org/content/339/6119/557>.
- P. C. Maurer, G. Kucsko, C. Latta, L. Jiang, N. Y. Yao, S. D. Bennett, F. Pastawski, D. Hunger, N. Chisholm, M. Markham, D. J. Twitchen, J. I. Cirac, and M. D. Lukin. Room-Temperature Quantum Bit Memory Exceeding One Second. *Science*, 336(6086):1283–1286, June 2012. ISSN 0036-8075, 1095-9203. doi: 10.1126/science.1220513. URL <http://www.sciencemag.org/content/336/6086/1283>.
- P. Neumann, J. Beck, M. Steiner, F. Rempp, H. Fedder, P. R. Hemmer, J. Wrachtrup, and F. Jelezko. Single-Shot Readout of a Single Nuclear Spin. *Science*, 329(5991):542–544, 2010. ISSN 0036-8075, 1095-9203. doi: 10.1126/science.1189075. URL <http://www.sciencemag.org/cgi/doi/10.1126/science.1189075>.

- M. Pfender, N. Aslam, H. Sumiya, S. Onoda, P. Neumann, J. Isoya, C. Meriles, and J. Wrachtrup. Nonvolatile quantum memory enables sensor unlimited nanoscale spectroscopy of finite quantum systems. *arXiv:1610.05675 [cond-mat, physics:quant-ph]*, Oct. 2016. URL <http://arxiv.org/abs/1610.05675>. arXiv: 1610.05675.
- T. Rosskopf, J. Zopes, J. M. Boss, and C. L. Degen. A quantum spectrum analyzer enhanced by a nuclear spin memory. *arXiv:1610.03253 [cond-mat, physics:quant-ph]*, Oct. 2016. URL <http://arxiv.org/abs/1610.03253>.
- T. Staudacher, F. Shi, S. Pezzagna, J. Meijer, J. Du, C. A. Meriles, F. Reinhard, and J. Wrachtrup. Nuclear Magnetic Resonance Spectroscopy on a (5-Nanometer)³ Sample Volume. *Science*, 339(6119):561–563, Feb. 2013. ISSN 0036-8075, 1095-9203. doi: 10.1126/science.1231675. URL <http://www.sciencemag.org/content/339/6119/561>.
- T. Staudacher, N. Raatz, S. Pezzagna, J. Meijer, F. Reinhard, C. A. Meriles, and J. Wrachtrup. Probing molecular dynamics at the nanoscale via an individual paramagnetic centre. *Nat Commun*, 6:8527, 2015. doi: 10.1038/ncomms9527. URL <http://www.nature.com/ncomms/2015/151012/ncomms9527/full/ncomms9527.html>.
- G. Waldherr, J. Beck, M. Steiner, P. Neumann, A. Gali, T. Frauenheim, F. Jelezko, and J. Wrachtrup. Dark States of Single Nitrogen-Vacancy Centers in Diamond Unraveled by Single Shot NMR. *Phys. Rev. Lett.*, 106(15):157601, Apr. 2011. ISSN 0031-9007, 1079-7114. doi: 10.1103/PhysRevLett.106.157601. URL <http://link.aps.org/doi/10.1103/PhysRevLett.106.157601>.
- S. Zaiser, T. Rendler, I. Jakobi, T. Wolf, S.-Y. Lee, S. Wagner, V. Bergholm, T. Schulte-Herbrüggen, P. Neumann, and J. Wrachtrup. Enhancing quantum sensing sensitivity by a quantum memory. *Nat. Commun.*, 7:12279, Aug. 2016. ISSN 2041-1723. doi: 10.1038/ncomms12279. URL <http://www.nature.com/doi/10.1038/ncomms12279>.

Reviewers' Comments:

Reviewer #2:

Remarks to the Author:

I am satisfied with the authors' changes to the manuscript and recommend publication of this excellent work.

Reviewer #3:

Remarks to the Author:

In my opinion the authors have thoroughly addressed the reviewers comments and I recommend the manuscript for publication in Nature Communications.